# Biosynthetic pathway for furanosteroid demethoxyviridin and identification of an unusual pregnane side-chain cleavage

Gao-Qian Wang[1], Guo-Dong Chen[1], Sheng-Ying Qin[2], Dan Hu [1], Takayoshi Awakawa[3], Shao-Yang Li[1], Jian-Ming Lv[1], Chuan-Xi Wang [1], Xin-Sheng Yao[1], Ikuro Abe [3] & Hao Gao [1]

Furanosteroids, represented by wortmannin, viridin, and demethoxyviridin, are a special group of fungal-derived, highly oxygenated steroids featured by an extra furan ring. They are well-known nanomolar-potency inhibitors of phosphatidylinositol 3-kinase and widely used in biological studies. Despite their importance, the biosyntheses of these molecules are poorly understood. Here, we report the identification of the biosynthetic gene cluster for demethoxyviridin, consisting of 19 genes, and among them 15 biosynthetic genes, including six cytochrome P450 monooxygenase genes, are deleted. As a result, 14 biosynthetic intermediates are isolated, and the biosynthetic pathway for demethoxyviridin is elucidated. Notably, the pregnane side-chain cleavage requires three enzymes: flavin-dependent Baeyer-Villiger monooxygenase, esterase, and dehydrogenase, in sharp contrast to the single cytochrome P450-mediated process in mammalian cells. Structure–activity analyses of these obtained biosynthetic intermediates reveal that the 3-keto group, the C1β–OH, and the aromatic ring C are important for the inhibition of phosphatidylinositol 3-kinase.

[1] Institute of Traditional Chinese Medicine and Natural Products, College of Pharmacy/Guangdong Province Key Laboratory of Pharmacodynamic Constituents of TCM and New Drugs Research, Jinan University, 510632 Guangzhou, People's Republic of China. [2] Clinical Experimental Center, First Affiliated Hospital of Jinan University, 510630 Guangzhou, People's Republic of China. [3] Graduate School of Pharmaceutical Sciences, The University of Tokyo, 7-3-1 HongoBunkyo-kuTokyo 113-0033, Japan. These authors contributed equally: Gao-Qian Wang, Guo-Dong Chen, Sheng-Ying Qin. Correspondence and requests for materials should be addressed to D.H. (email: thudan@jnu.edu.cn) or to I.A. (email: abei@mol.f.u-tokyo.ac.jp) or to H.G. (email: tghao@jnu.edu.cn)

Steroids are modified triterpenoids containing the tetracyclic system of lanosterol, but lacking the three methyl groups at C4 and C14. Further modifications in the side chain lead to different sub-classes of steroids bearing $C_{18}$–$C_{29}$ skeletons. They are some of the most widely distributed small molecules in nature and serve a myriad of biological functions. Sterols are the most important form of steroids, with a hydroxyl group at C3 and a skeleton derived from cholestane, among which cholesterol in animals, sitosterol in plants, and ergosterol in fungi are well-known molecules, as they are the essential components of the cellular membranes in these eukaryotic organisms[1, 2]. In addition, sterols are important precursors for many biologically important molecules, such as the steroid hormones from animals and the cardenolides from plants, by extensive carbon degradation[3, 4]. In fungi, the oxidative removal of carbons from sterol precursors also produces active molecules, such as wortmannin, viridin, and demethoxyviridin (**1**), which are called furanosteroids because all of these molecules contain an extra furan ring fused between C4 and C6 of the steroidal framework (Fig. 1a)[5]. Since viridin was first discovered in 1945[6], extensive biological studies of this class of compounds have been performed, which revealed that furanosteroids possess a variety of important biological properties, including antifungal, anti-inflammatory, and antibacterial activities[7, 8]. Especially, furanosteroids are nanomolar-potency inhibitors of phosphatidylinositol 3-kinase (PI3K), among which wortmannin has been developed as a commercial PI3K inhibitor widely used in various biological studies[9, 10]. Notably, a semi-synthetic analog of wortmannin, PX-866, was tested in a phase II clinical trial for treating cancers[11]. The intriguing structures and excellent biological activities of furanosteroids have thus led to extensive efforts toward their total chemical synthesis over the past 20 years, and the stereoselective synthesis of wortmannin and (−)-viridin was finally achieved in 2017[12, 13]. However, as compared with the progress in chemical synthesis, the biosynthesis of these important molecules in fungi is poorly understood.

Early isotope labeling experiments with [13]C-labeled mevalonate or acetate revealed that viridin, demethoxyviridin, and wortmannin are formed from two farnesyl units in a steroid-like manner, and do not originate from a diterpenoid[14–16]. A subsequent study using [14]C-labeled lanosterol confirmed that it is converted to viridin via a normal steroidal pathway[17]. To gain further insight into the biosynthesis of demethoxyviridin from lanosterol, Hanson et al. identified a group of ergosterol side-chain-derived C6 and C7 alcohols sharing a common origin with demethoxyviridin. This suggested that the removal of the sterol side chain during demethoxyviridin biosynthesis occurs between C20 and C22, in a pathway analogous to the sterol side-chain cleavage during the biosynthesis of steroid hormones in mammalian cells[18, 19]. Despite these advances, the genetic basis for the biosynthesis of furanosteroids has remained elusive. Recently, a transcriptional comparison of *Trichoderma virens* IMI 304061, a high producer of viridin, and its mutant strain deficient in secondary metabolite production identified a four-gene cluster predicted to be responsible for the biosynthesis of viridin;[20] however, it was soon realized that this gene cluster is involved in the biosynthesis of volatile terpene compounds, rather than viridin[21]. During our manuscript submission, Bansal et al. reported a biosynthetic gene cluster for viridin, but they did not provide substantial evidence for the biosynthetic pathway of viridin[22].

In our previous explorations for bioactive secondary metabolites from fungi[23, 24], we identified the endolichenic fungus *Nodulisporium* sp. (no. 65-12-7-1), which can produce large amounts of demethoxyviridin (**1**) and several analogs[25, 26]. These findings provided a good chance to elucidate its biosynthesis.

Here, we report the identification of the gene cluster and the biosynthetic pathway for **1**, by the combinational use of a transcriptome comparison analysis, CRISPR-Cas9-based gene disruption, an *Aspergillus oryzae* NSAR1 heterologous gene expression system, and an in vitro enzymatic assay. Our study sets the stage to uncover the biosyntheses of other furanosteroids and expands the chemical diversity of pharmaceutically important furanosteroids by engineered biosyntheses.

## Results

**Identification of the gene cluster for demethoxyviridin.** Although in most cases, terpene cyclase is often clustered with its downstream modification enzymes in fungal genomes[27], the lanosterol-derived triterpenes/steroids, including ergosterol and ganoderic acid, do not conform to this rule[28, 29]. This could be the reason why Kenerley et al. failed to find the biosynthetic gene cluster of viridin, when using terpene cyclase as the starting point[21]. Therefore, a different strategy based on other features of

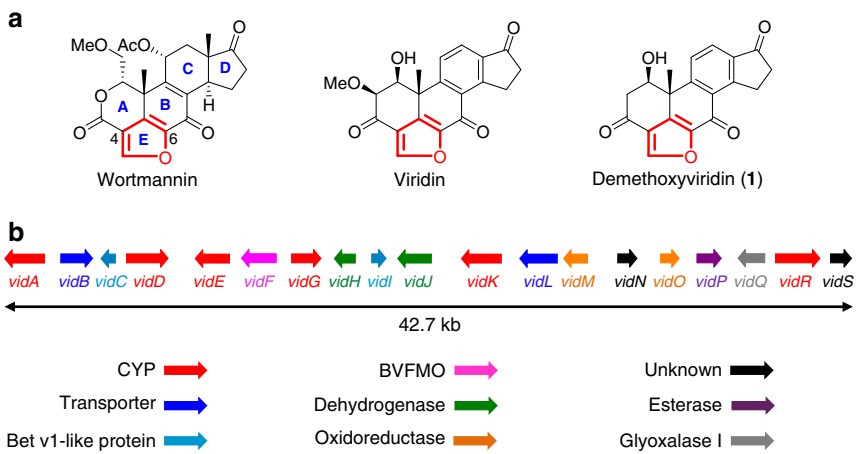

**Fig. 1** Representative furanosteroids and biosynthetic gene cluster of demethoxyviridin (**1**). **a** Structures of wortmannin, viridin, and demethoxyviridin (**1**). **b** Gene map of the demethoxyviridin biosynthetic gene cluster from *Nodulisporium* sp. (no. 65-12-7-1), consisting of 19 genes from *vidA* (*g3266*) to *vidS* (*g3284*). The arrow indicates the direction from the start to the stop codon. Different types of genes are indicated by different colors and among them red arrows indicate six CYP genes

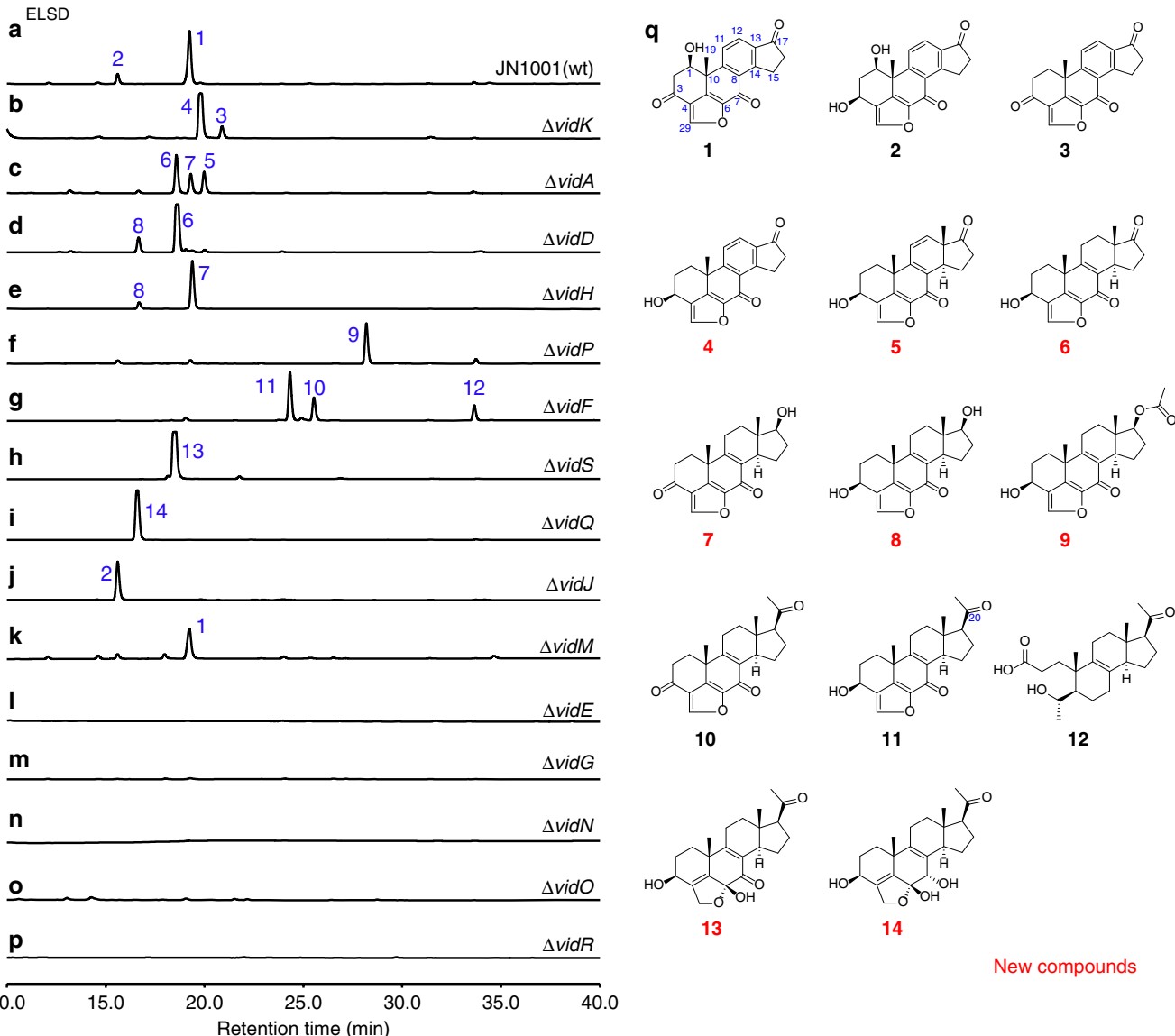

**Fig. 2** Metabolite analysis of the *vid* gene deletion mutants. **a–p** HPLC profiles of extracts from the *vid* gene deletion mutants. **q** Structures of the compounds isolated from the *vid* gene deletion mutants. The chromatograms were monitored with an evaporative light scattering detector (ELSD). The new compounds are indicated by red numbers

furanosteroids should be adopted. Since demethoxyviridin possesses a highly oxygenated structure, we inferred that there should be multiple cytochrome P450 monooxygenase (CYP) genes in its biosynthetic gene cluster. Thus, the CYP gene clusters in the genome could be potential targets. To identify the biosynthetic gene cluster of demethoxyviridin, we sequenced the whole genome of *Nodulisporium* sp. (no. 65-17-2-1), and identified a total of 103 CYP genes in the genome. Analyses of the relative localizations of these genes in the genome revealed 12 CYP clusters (clusters I–XII) containing two or more CYP genes (Supplementary Fig. 1). To determine the candidate gene cluster of demethoxyviridin, one CYP gene from each gene cluster was randomly selected (Supplementary Table 1), and its expression was analyzed by reverse transcription PCR (RT-PCR), under demethoxyviridin productive (maltose medium) and non-productive (Czapek medium) conditions (Supplementary Fig. 2). As shown in Supplementary Fig. 3, in only three clusters, II, V, and IX, the selected genes were upregulated under the demethoxyviridin productive conditions. To further narrow the list, we investigated the expression of the other genes in these

three clusters, and found that only in cluster V, all of the genes including six CYP genes were upregulated under the demethoxyviridin productive conditions (Supplementary Fig. 4). This result was further confirmed by RNA sequencing (RNA-Seq), and the genes from *g3262* to *g3284* were found to be upregulated, except for the non-expressed gene, *g3263* (Supplementary Fig. 5 and Supplementary Table 5).

To experimentally validate whether cluster V is responsible for the biosynthesis of demethoxyviridin, we recently established an efficient CRISPR-Cas9-based gene disruption method in fungi by simultaneously introducing the in vitro transcriptional gRNA and the linear marker gene cassette into a Cas9-expressing strain[30]. We then used this method to disrupt the CYP gene *g3266* (*vidA*) in the cluster, to generate the mutant strain Δ*vidA*-JN1001. As compared with the parent strain JN1001, in which **1** and its 3-keto reduced derivative demethoxyviridiol (**2**) were abundantly produced, the Δ*vidA*-JN1001 mutant strain lost the ability to produce **1** and **2**, but instead produced three additional compounds, **5** (2.6 mg L⁻¹), **6** (3.4 mg L⁻¹), and **7** (2.1 mg L⁻¹) (Fig. 2c). This result clearly demonstrated that cluster V is

**Table 1 Putative functions of genes in the *vid* cluster**

| Gene | Amino acids (base pairs) | Protein homolog [origin] | Similarity/identity (%) | Proposed function |
|---|---|---|---|---|
| *vidA* | 580 (1801) | ABE60729.1 [Trichoderma virens] | 74/56 | Cytochrome P450 monooxygenase |
| *vidB* | 486 (1534) | XP_003720468.1 [Magnaporthe oryzae 70-15] | 78/65 | Membrane transportor |
| *vidC* | 156 (591) | OAL05076.1 [Stagonospora sp. SRC1lsM3a] | 43/29 | Bet v1-like protein |
| *vidD* | 601 (1869) | ABE60729.1 [Trichoderma virens] | 76/58 | Cytochrome P450 monooxygenase |
| *vidE* | 556 (1671) | CRG92635.1 [Talaromyces islandicus] | 65/48 | 25-Hydroxycholesterol 7-alpha-hydroxylase |
| *vidF* | 541 (1710) | XP_018005652.1 [Phialophora attae] | 70/51 | Baeyer-Villiger monooxygenase |
| *vidG* | 490 (1473) | GAP90667.1 [Rosellinia necatrix] | 86/73 | Cytochrome P450 monooxygenase |
| *vidH* | 283 (852) | OGM39379.1 [Aspergillus bombycis] | 59/44 | Gluconate 5-dehydrogenase |
| *vidI* | 154 (601) | OAL05076.1 [Stagonospora sp. SRC1lsM3a] | 74/50 | Bet v1-like protein |
| *vidJ* | 505 (1576) | OKP10358.1 [Penicillium subrubescens] | 68/55 | 6-Hydroxy-D-nicotine oxidase |
| *vidK* | 578 (1840) | ABE60729.1 [Trichoderma virens] | 75/60 | Cytochrome P450 monooxygenase |
| *vidL* | 598 (1797) | XP_008601859.1 [Beauveria bassiana ARSEF 2860] | 71/55 | Major facilitator superfamily transporter |
| *vidM* | 350 (1053) | OCW38945.1 [Diaporthe helianthi] | 48/29 | C-3 sterol dehydrogenase/C-4 decarboxylase |
| *vidN* | 254 (894) | XP_003720470.1 [Magnaporthe oryzae 70-15] | 67/49 | Hypothetical protein MGG_10346 |
| *vidO* | 253 (876) | KKY14473.1 [Diplodia seriata] | 67/52 | Short chain oxidoreductase |
| *vidP* | 354 (1065) | XP_001260764.1 [Aspergillus fischeri NRRL 181] | 64/51 | Arylesterase |
| *vidQ* | 310 (1150) | XP_003857325.1 [Zymoseptoria tritici IPO323] | 71/56 | Monomeric glyoxalase I |
| *vidR* | 547 (2041) | ELQ58335.1 [Magnaporthe oryzae P131] | 86/74 | Cytochrome P450 monooxygenase |
| *vidS* | 263 (853) | XP_012746650.1 [Pseudogymnoascus destructans 20631-21] | 84/76 | Hypothetical protein GMDG_08190 |

responsible for the biosynthesis of demethoxyviridin. In order to determine the exact boundary of the cluster, the genes on both ends of the cluster V (*g3262–g3284*) were individually disrupted. As shown in Supplementary Fig. 6, the inactivations of the upstream *g3262* (an alcohol dehydrogenase), *g3264* (a hypothetical protein), and *g3265* (a transporter protein), and the downstream *g3285* (a heterokaryon incompatibility protein) had no effects on the production of **1** and **2**, whereas the disruption of either *g3266* (*vidA*) or *g3284* (*vidS*) abolished their production and led to the production of different compounds (Fig. 2c, h). These results provided strong evidence that we had correctly identified the biosynthetic gene cluster of demethoxyviridin, which consists of 19 genes from *g3266* (*vidA*) to *g3284* (*vidS*), including 15 biosynthetic genes for the structural assembly of demethoxyviridin, two transporter genes, and two Bet v1-like genes (Fig. 1b and Table 1). For convenience, we designated the identified demethoxyviridin gene cluster as the *vid* cluster, and renamed the 19 genes (*vidA–vidS*) (Fig. 1b and Table 1).

**Deciphering the biosynthetic pathway for demethoxyviridin.** To explicitly characterize the function of each gene in the *vid* cluster and decipher the biosynthetic pathway of demethoxyviridin, 15 biosynthetic genes, including the six CYP genes, were disrupted and the resulting mutant strains were subjected to a metabolite analysis (Fig. 2).

VidK catalyzes the final step of demethoxyviridin biosynthesis via C1β hydroxylation. The *vidK* gene encodes a CYP enzyme. When it was deleted, the resulting mutant strain, ΔvidK-JN1001, lost the ability to produce **1** and **2** but produced two additional compounds, **3** (3.5 mg L$^{-1}$) and **4** (10.5 mg L$^{-1}$) (Fig. 2b). Both **3** and **4** were structurally determined to be the 1-dehydroxylated **1** and **2**, respectively (Fig. 2q). These results suggested that VidK catalyzes the final step in the biosynthesis of demethoxyviridin, by attaching a β-hydroxyl group at the C1 position (Fig. 3).

VidA and VidD account for the aromatization of ring C. The *vidA* and *vidD* genes encode two other CYPs in the cluster. When *vidA* was disrupted, the resulting mutant strain ΔvidA-JN1001

failed to produce compounds **1** and **2**, but accumulated three other compounds, **5** (2.6 mg L$^{-1}$), **6** (3.4 mg L$^{-1}$), and **7** (2.1 mg L$^{-1}$) (Fig. 2c). These compounds were structurally determined via a rigorous NMR investigation (Fig. 2q). Structural comparisons of these compounds revealed that compound **5** has just one more methyl group at C13 than compound **4**, produced by ΔvidK-JN1001, suggesting that VidA is responsible for the removal of the C18 methyl group before the action of VidK, a process analogous to the removal of the methyl group at C10 during estrogen biosynthesis[31] (Fig. 3). This conclusion is also supported by the fact that compounds **5–7** isolated from ΔvidA-JN1001 all bear a C18 methyl group (Fig. 2q). Similarly, when we disrupted *vidD*, the resulting mutant strain ΔvidD-JN1001 also generated two distinctive compounds, **6** (6.5 mg L$^{-1}$) and **8** (4.5 mg L$^{-1}$) (Fig. 2d). After structure determination, we found that **6** lacks a double bond between C11 and C12, as compared with **5** generated by ΔvidA-JN1001 (Fig. 2q), suggesting that VidD is involved in the dehydrogenation of C11 and C12 before the VidA-mediated oxidative demethylation (Fig. 3). Consistent with this conclusion, **8** also lacks a double bond between C11 and C12 (Fig. 2q). These results clearly indicate that the aromatization of ring C of the furanosteroids is accomplished by the two CYP monooxygenases, VidA and VidD.

VidH, VidP, and VidF catalyze the pregnane side-chain cleavage. The *vidH* gene encodes an NAD(P)-binding dehydrogenase. The disruption of *vidH* resulted in the production of two distinct compounds, **7** (5.5 mg L$^{-1}$) and **8** (3.5 mg L$^{-1}$) (Fig. 2e). As compared with **6**, produced by ΔvidD-JN1001, both **7** and **8** carry a 17β-hydroxyl group, rather than the 17-keto in **6** (Fig. 2q), suggesting that VidH catalyzes the dehydrogenation of the 17-OH to form the 17-keto, before the VidD-mediated reaction. The *vidP* gene encodes an esterase. The disruption of *vidP* generated a single compound, **9** (9.5 mg L$^{-1}$) (Fig. 2f), which was determined to the 17-O-acetylated product of **8** (Fig. 2q), indicating that VidP is responsible for the hydrolysis of the 17-O-acetyl side chain, followed by the VidH-mediated dehydrogenation (Fig. 3). In contrast, *vidF* encodes a flavin-dependent Baeyer-Villiger monooxygenase (BVFMO). The disruption of *vidF* thus led to the

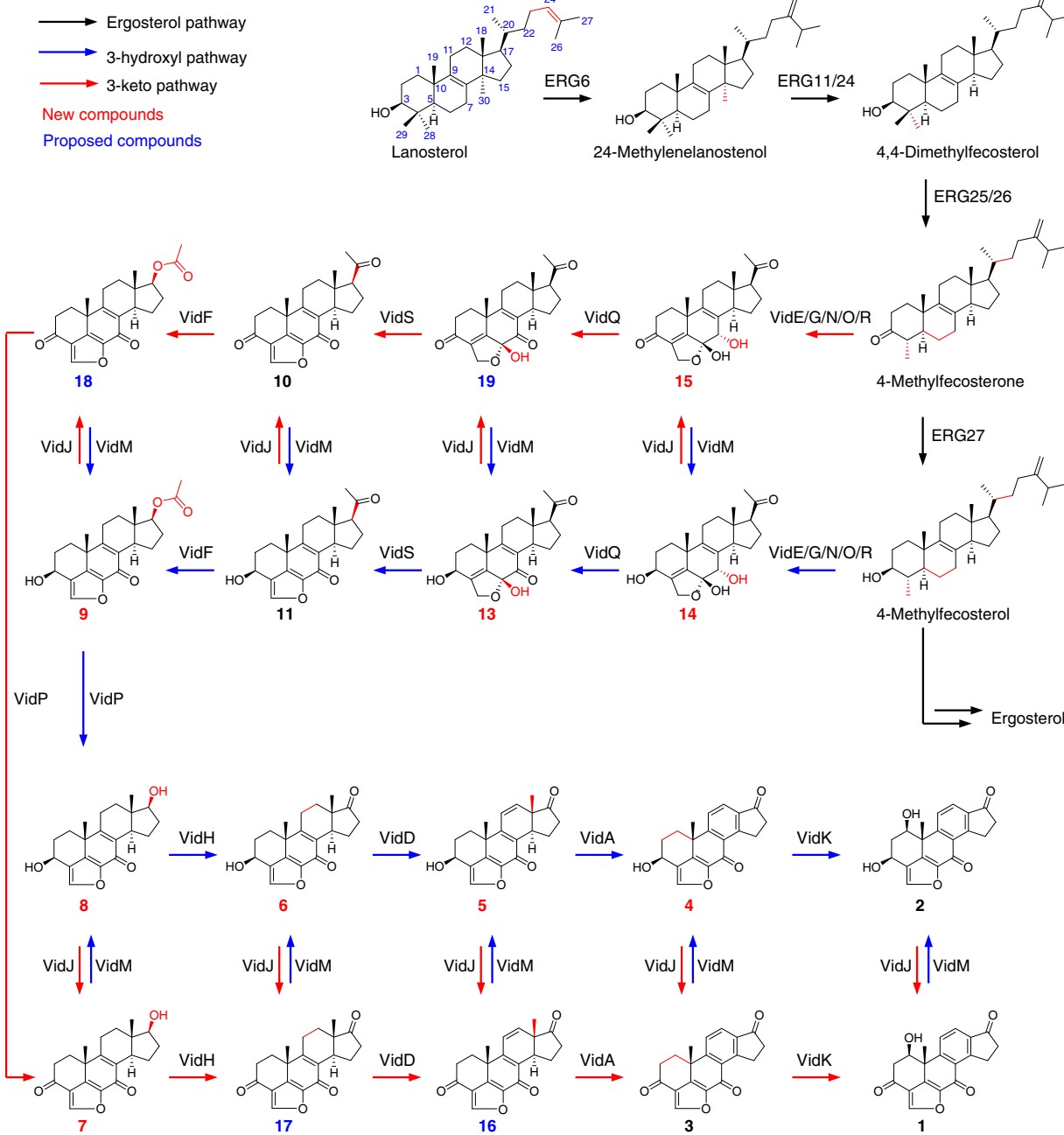

**Fig. 3** Complete biosynthetic pathway of demethoxyviridin (**1**). Proposed biosynthetic pathway for **1** (marked by red arrows), and its 3-OH derivative **2** (marked by blue arrows). The ergosterol pathway is marked by black arrows. New compounds and proposed compounds are indicated by red and blue numbers, respectively. The enzyme-mediated modification sites in individual reaction are marked by red

production of three additional compounds, **10** (2.2 mg L$^{-1}$), **11** (3.2 mg L$^{-1}$), and **12** (1.7 mg L$^{-1}$) (Fig. 2g). Their structures were determined by NMR, which revealed that **11** only lacks an oxygen atom between C17 and C20, as compared to **9** generated by Δ*vidP*-JN1001 (Fig. 2q), suggesting that VidF is involved in the Baeyer–Villiger oxidation of the C-20-keto intermediate **11**, to afford the 17-*O*-acetylated ester **9** before hydrolysis by VidP (Fig. 3). The above results indicate that during the biosynthesis of demethoxyviridin, three enzymes (BVFMO, esterase, and dehydrogenase) are required for the cleavage of the pregnane side chain, in sharp contrast to the single CYP-catalyzed progesterone side-chain cleavage in mammalian cells[32].

Furan ring maturation by VidS followed by VidQ-mediated dehydrogenation. The furan ring fused between C4 and C6 is the

most unique part in the furanosteroids, and its formation has long remained enigmatic. VidS is annotated as a hypothetical protein. When it was deleted, the resulting mutant strain Δ*vidS*-JN1001 produced a single metabolite, **13** (12.6 mg L$^{-1}$) (Fig. 2h). The structural assignment of **13** via NMR and X-ray crystallographic analyses revealed that it possesses a very similar structure to **11**, but differs in the furan ring, which contains an unusual pentacyclic hemiacetal (Fig. 2q). These results suggested that VidS is involved in the dehydration of a cyclic hemiacetal to form a mature furan ring, possibly via 1,4 dehydration. The *vidQ* gene encodes a protein that is highly homologous to the Zn$^{2+}$-dependent glyoxalase I, which is known to catalyze the isomerization of the hemithioacetal, formed spontaneously from alpha-oxoaldehyde and glutathione, to *S*-2-

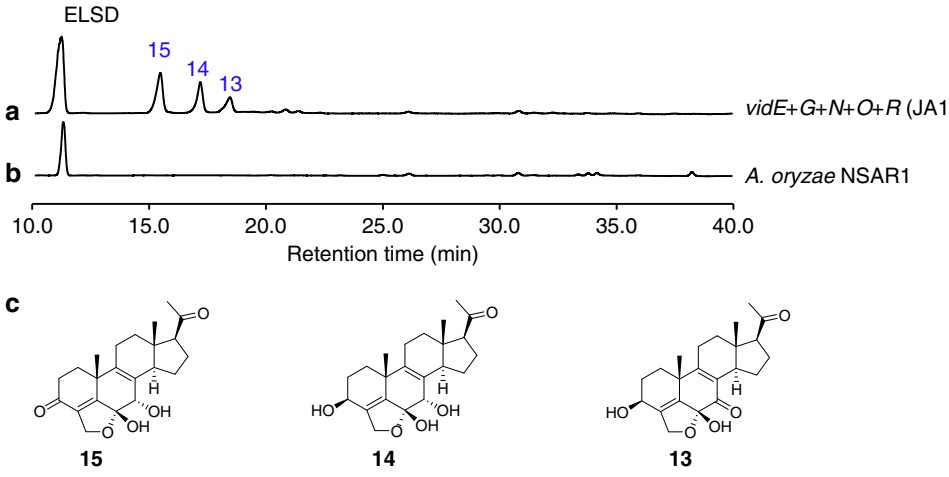

**Fig. 4** HPLC analysis of culture extract of JA1. **a** *A. oryzae* harboring *vidE*, *vidG*, *vidR*, *vidN*, and *vidO* (JA1). **b** *A. oryzae*. **c** Structures of compounds **13**–**15**

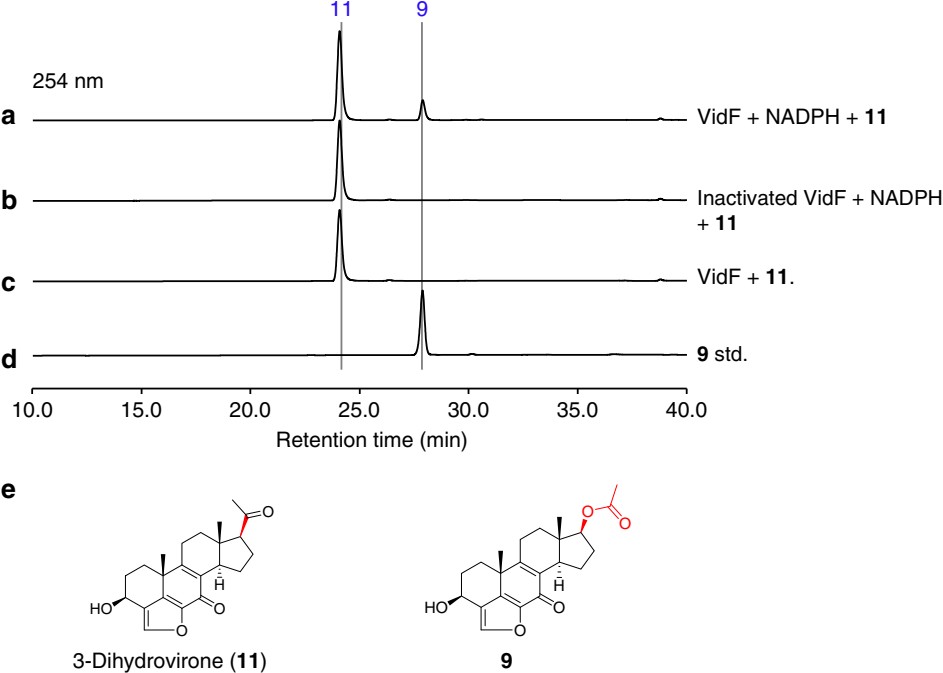

**Fig. 5** HPLC analysis of in vitro enzymatic assay. **a** **11** with VidF and NADPH. **b** **11** with inactivated VidF and NADPH. **c** **11** with VidF. **d** Standard (std.) of **9**. **e** Structures of compounds **9** and **11**. The enzyme-mediated modification sites in individual reaction are marked by red

hydroxyacylglutathione derivatives[33]. Inactivation of *vidQ* led to the production of the major compound **14** (14.2 mg L$^{-1}$) (Fig. 2i). An HRESIMS analysis showed that **14** is 2 Da larger than **13**, suggesting that **14** is the reduced product of **13**. A subsequent NMR analysis revealed that **14** carries a hydroxyl group at C7, rather than the keto group in **13** (Fig. 2q). These results indicated that VidQ is involved in the dehydrogenation of the 7-OH to form the 7-keto group during demethoxyviridin biosynthesis (Fig. 3).

VidJ and VidM are responsible for the mutual conversion between the 3-OH and 3-keto groups. Since the 3-keto and 3-hydroxyl metabolites often coexist, there might be some enzymes involved in this mutual transformation. The *vidJ* and *vidM* genes encode a 6-hydroxy-D-nicotine oxidase and a C-3 sterol dehydrogenase, respectively. The disruption of *vidJ* predominantly produced the 3-OH derivative **2** (Fig. 2j), while the deletion of *vidM* generated the 3-keto form **1** (Fig. 2k). Based on these results, we concluded that *vidJ* is likely to be responsible for

the oxidation of the 3-OH to the 3-keto, while *vidM* is involved in the opposite reaction to form the 3-OH derivatives throughout the biosynthesis of demethoxyviridin (Fig. 3).

VidE, VidG, VidR, VidN, and VidO are involved in the early stages of demethoxyviridin biosynthesis. Except for the two Bet v1-like genes (*vidC* and *vidI*) and the two transporter genes (*vidB* and *vidL*), only five biosynthetic genes in the gene cluster remained to be characterized, including three CYP genes (*vidE*, *vidG*, and *vidR*), one short chain oxidoreductase gene (*vidO*), and one hypothetical gene (*vidN*). The deletion of these genes abolished **1** and **2**, but did not yield stable intermediates (Fig. 2l–p). According to the knowledge obtained from the above deletion experiments, we could infer that these five genes are involved in the early biosynthetic pathway of demethoxyviridin, including the sterol side-chain cleavage between C20 and C22 and the furan ring skeleton formation, to form the intermediate **14** or its 3-keto derivative **15** (Fig. 3). To verify this hypothesis, we constructed an *A. oryzae* NSAR1[34] transformant strain JA1

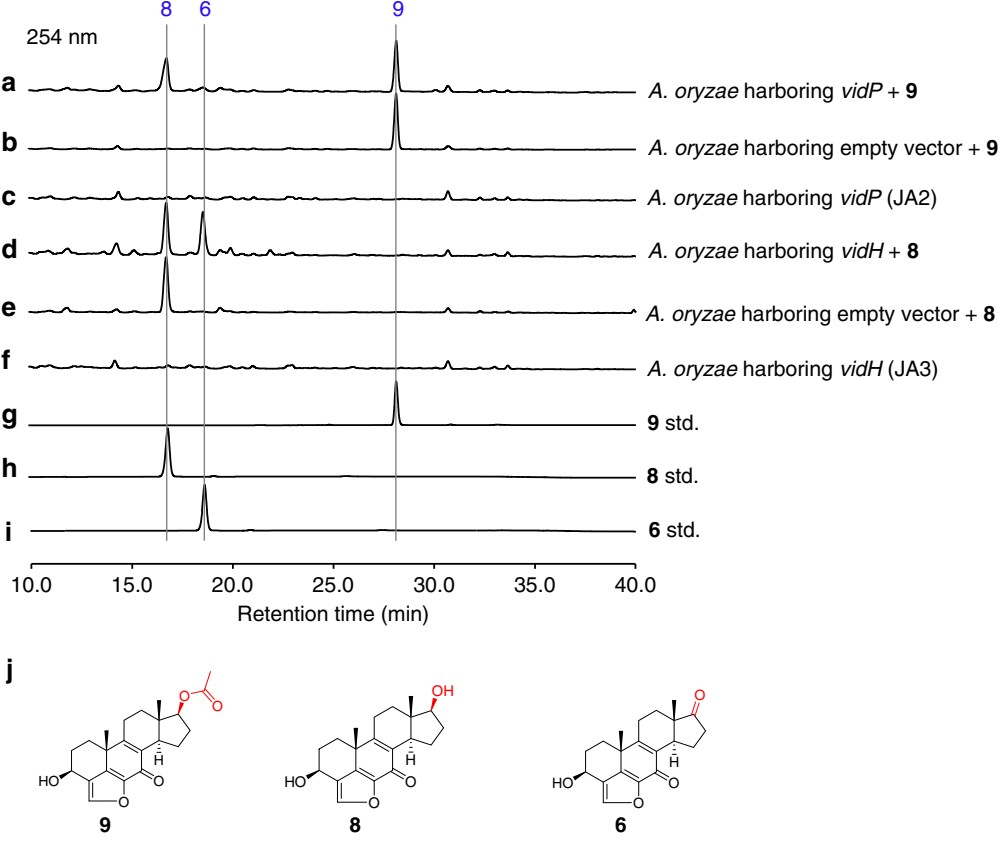

**Fig. 6** HPLC analysis of feeding experiments. **a** *A. oryzae* harboring *vidP* (JA2) with **9**. **b** *A. oryzae* harboring empty vector with **9**. **c** *A. oryzae* harboring *vidP*. **d** *A. oryzae* harboring *vidH* (JA3) with **8**. **e** *A. oryzae* harboring empty vector with **8**. **f** *A. oryzae* harboring *vidH*. **g** Standard (std.) of **9**. **h** Standard (std.) of **8**. **i** Standard (std.) of **6**. **j** Structures of compounds **9**, **8**, and **6**. The enzyme-mediated modification sites in individual reaction are marked by red

co-expressing all five genes (*vidE*, *vidG*, *vidR*, *vidN*, and *vidO*). After JA1 was cultured in the induction medium for 5 days, the mycelia extract was analyzed by HPLC. As a result, three additional compounds, **13** (1.1 mg L$^{-1}$), **14** (1.4 mg L$^{-1}$), and **15** (1.7 mg L$^{-1}$), were detected in the transformant JA1 (Fig. 4a), but not in the control strain (Fig. 4b). These compounds were isolated and structurally determined. The fact that the five-gene co-expressing strain indeed produces the key intermediate **14** and its 3-keto derivative **15** conceivably demonstrated that these five genes are involved in the early stage of demethoxyviridin biosynthesis, as illustrated in Fig. 3.

**Unusual pregnane side-chain cleavage by VidF, VidP, and VidH**. The above deletion experiments revealed the unusual pregnane side-chain cleavage by the collaboration of three enzymes: VidF (BVFMO), VidP (esterase), and VidH (dehydrogenase), during the demethoxyviridin biosynthesis, which is distinct from the single CYP-catalyzed process during the biosynthesis of steroid hormones in mammalian cells (Supplementary Fig. 7). To further confirm this process, an in vitro enzymatic assay using purified recombinant VidF, obtained from an *Escherichia coli* expression system, was performed (Supplementary Fig. 8). Since VidF shares high similarity with the members of the Type I BVFMOs, an in vitro reaction was performed in the presence of NADPH and FAD. As expected, VidF converted **11** to **9**, by inserting an oxygen atom between C17 and C20 (Fig. 5a, b). The removal of NADPH from the reaction abolished the conversion (Fig. 5c). These results confirmed the role of VidF in the initiation of pregnane side-chain cleavage. Since the VidP and VidH proteins could not be expressed in *E. coli*, feeding experiments using *A. oryzae* NSAR1 strains expressing either *vidP* (JA2)

or *vidH* (JA3) were performed, to verify their functions. As a result, JA2 converted **9** into **8**, while the control strain did not (Fig. 6a, b), and JA3 harboring *vidH* with **8** generated **6** (Fig. 6d, e). These results clearly established that the cleavage of C17−C20 during demethoxyviridin biosynthesis is catalyzed by the three enzymes, VidF, VidP, and VidH.

The pregnane side-chain cleavage of progesterone (**20**) by fungi has been well characterized, and it is mediated by a BVFMO, esterase, and dehydrogenase. However, the biochemical basis for this process is still not clear[35]. Since the cleavage between C17 and C20 by VidF, VidP, and VidH during the biosynthesis of demethoxyviridin is quite similar to that reported in the cleavage of progesterone[35], we hypothesized that the side-chain cleavages of demethoxyviridin and progesterone are likely to be mediated by homologous enzymes. To verify this hypothesis, *vidF*, *vidH*, and *vidP* were simultaneously transformed into *A. oryzae* NSAR1 to construct the three-gene co-expression strain, JA4. As expected, incubating JA4 with progesterone (**20**) successfully led to the production of four additional compounds, **21**−**24** (Fig. 7). These results are consistent with previous reports about the fungal degradation of progesterone[35].

**Structure−activity analysis of demethoxyviridin derivatives**. In the course of our work, we obtained 14 biosynthetic intermediates, including 8 previously unknown compounds via targeted gene deletions, which provided an opportunity to study the structure−activity relationships of demethoxyviridin, as these compounds appear in pairs that differ at only one position. We evaluated the inhibitory effects of all of these compounds on PI3K, using NVP-BEZ235 (BEZ235) as a positive control (Fig. 8). Among them, demethoxyviridin (**1**) showed the most potent

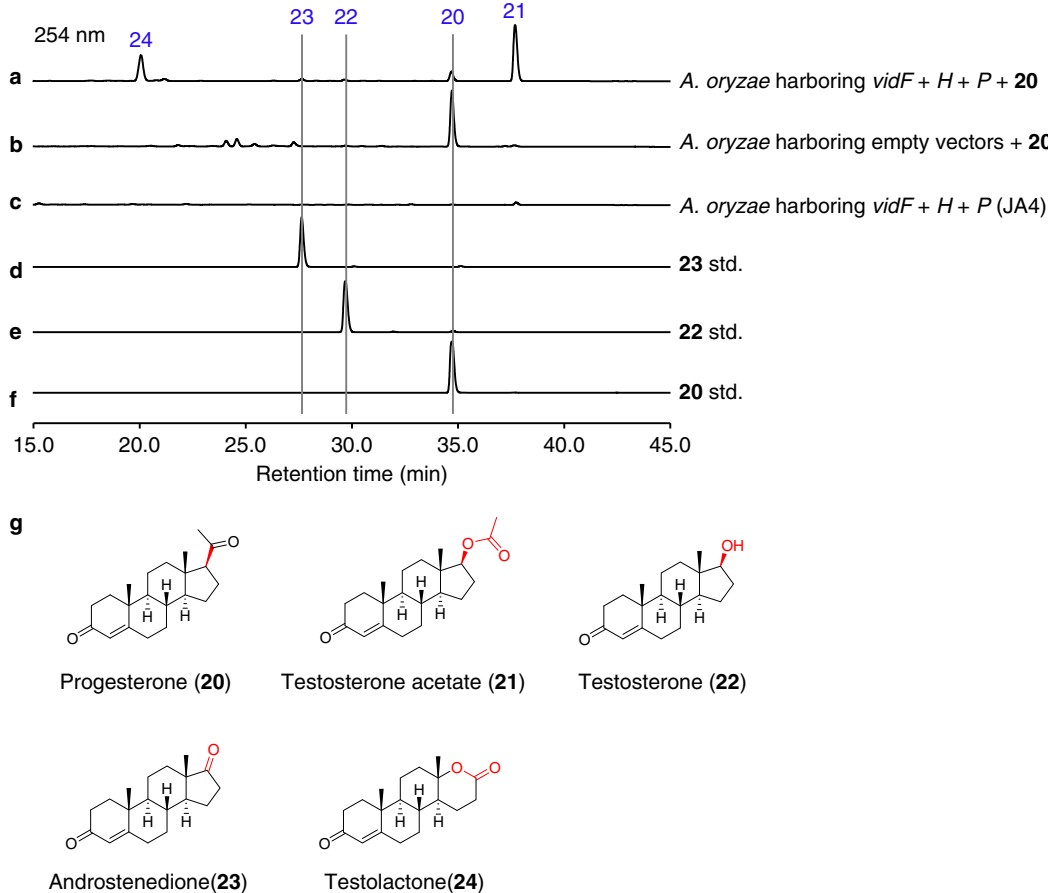

**Fig. 7** Progesterone side-chain cleavage by VidF, VidP, and VidH. **a** *A. oryzae* harboring *vidF*+*H*+*P* (JA4) with **20**. **b** *A. oryzae* harboring empty vectors with **20**. **c** *A. oryzae* harboring *vidF*+*H*+*P*. **d** Standard (std.) of **23**. **e** Standard (std.) of **22**. **f** Standard (std.) of **20**. **g** Structures of compounds **20**−**24**. The enzyme-mediated modification sites in individual reaction are marked by red

inhibitory activity against PI3K with an IC$_{50}$ value of 6.6 nM, which is much better than that of **2** (IC$_{50}$ = 1074 nM), suggesting that the 3-keto group is critical for the activity. This conclusion is also supported by the comparison of the activities between **3** and **4**, **7** and **8**, and **10** and **11** (Fig. 8a). A further comparison of **6** and **8** suggested that the 17β-OH is more important than the 17-keto, and the acetylation of the 17β-OH (**9**) abolishes the activity (Fig. 8a), consistent with a previous study on wortmannin[36]. Notably, we found that the β-OH at C-1 is important for inhibiting PI3K, as **1** and **2** exhibit more potent activities than their corresponding 1-dehydroxyl products **3** and **4**, respectively. In addition, compound **4**, with an aromatic ring C, shows higher inhibitory activity than compounds **5** and **6** without an aromatic ring C, indicating that the aromatization of ring C also contributes to the activity (Fig. 8a).

Since the PI3K pathway is frequently activated in a variety of human cancers, including breast cancer[37], inhibitors of this pathway are potential anti-cancer therapeutics. Therefore, we used the breast cancer cell line MCF-7 to evaluate the cytotoxicities of these compounds (Supplementary Fig. 9). Although **1** showed the most potent inhibitory activity toward PI3K in an in vitro kinase inhibition assay (Fig. 8), it only exhibited moderate cytotoxicity against MCF-7, perhaps due to its instability in PBS and culture media[38]. In contrast, our obtained compound **7** showed the most potent cytotoxicity against MCF-7, with an IC$_{50}$ value of 15.0 μM, followed by **2** (Supplementary Fig. 9). These results indicated that compound **7** is a more stable PI3K inhibitor.

## Discussion

Furanosteroids are a group of remarkable, highly oxygenated steroidal metabolites derived from fungi, and are nanomolar-potency inhibitors of PI3K. Here, we reported the identification of the biosynthetic gene cluster for the furanosteroid demethoxyviridin (**1**), and deciphered its biosynthetic pathway. Notably, we found that an unusual pregnane side-chain cleavage process occurs during demethoxyviridin biosynthesis, which is distinct from the biosyntheses of structure-related steroid hormones in mammalian cells. Moreover, 14 demethoxyviridin intermediates were obtained by gene deletion experiments, providing insights into the structure–activity relationships of furanosteroids for PI3K inhibition.

Although furanosteroids were discovered in fungi more than 70 years ago, their biosynthesis is poorly understood. Golder et al. first demonstrated that lanosterol is the biosynthetic precursor of furanosteroids[17]. Since lanosterol also serves as the precursor of ergosterol, it is conceivable that the biosynthesis of furanosteroids bifurcates from the ergosterol biosynthetic pathway. Although the branch point for the biosynthesis of demethoxyviridin from the ergosterol pathway is not well understood, Hanson et al. previously isolated a series of ergosterol side-chain-derived C6 and C7 alcohols, commonly originating with demethoxyviridin from *Nodulisporium hinnuleum*. This suggests that the 24-methylation is likely to occur at the early stage of ergosterol biosynthesis, as found in *A. fumigatus*[39]. The subsequent demethylation at C14, Δ$^{14}$-bond reduction, and removal of the first α-methyl group at C4 are shared by both the ergosterol and demethoxyviridin

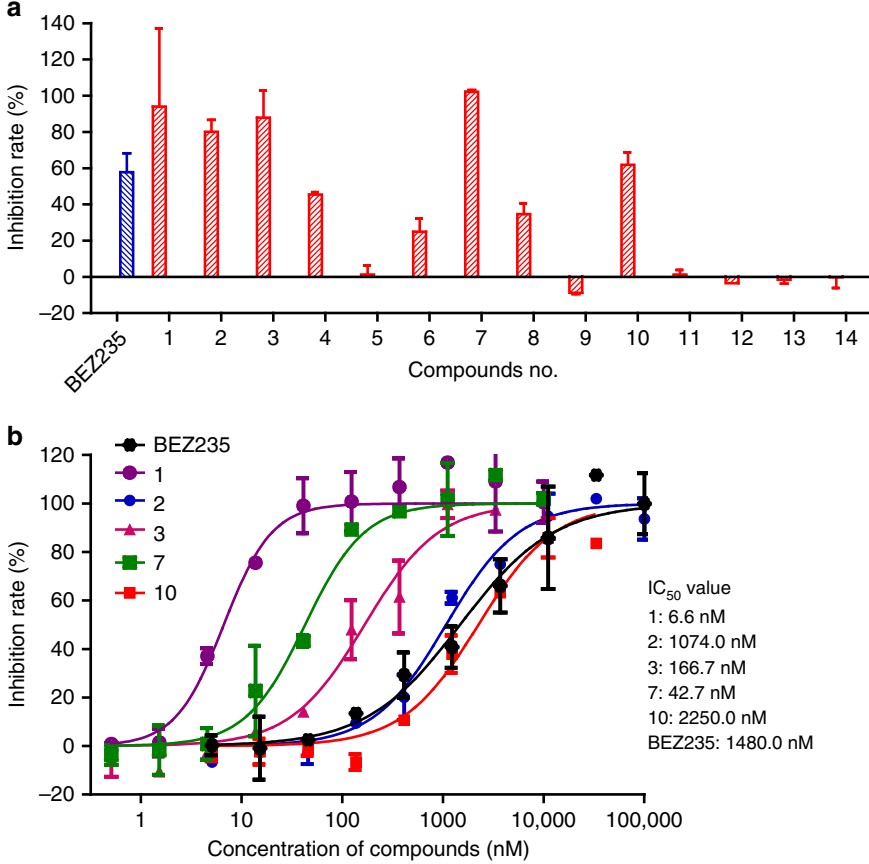

**Fig. 8** Phosphatidylinositol 3-kinase (PI3K) inhibitory activity. **a** The inhibitory effects of compounds **1**–**14** against PI3K, with BEZ235 as the positive control. The inhibition ratio of a certain sample on PI3K was calculated according to the following formula: PI3K inhibition (%) = (ER$_{sample}$ − ER$_{0\%}$)/(ER$_{100\%}$ − ER$_{0\%}$) × 100%, as described in method section. Data represent mean ± SD of two replicates. **b** The IC$_{50}$ values of compounds **1**–**3**, **7**, **10**, and BEZ235. The IC$_{50}$ values were estimated using the "Sigmoidal dose-response (variable slope)" equation in the GraphPad Prism® Version 5.0 software. Data represent mean ± SD of two replicates

biosynthetic pathways[40]. Thus, we hypothesized that the bifurcation from ergosterol to demethoxyviridin biosynthesis is likely to occur after the removal of the first α-methyl group at C4, and the resulting product either undergoes the second demethylation to ergosterol biosynthesis, or receives alternative modifications to trigger the demethoxyviridin biosynthesis (Fig. 3). Recently, we isolated a series of 4-methyl-progesteroid derivatives from *Nodulisporium* sp. (no. 65-12-7-1)[41], suggesting that the cleavage of the C20−C22 bond, to form the 20-keto, might be a key step in the divergence of furanosteroid biosynthesis from ergosterol biosynthesis. Based on these facts, we have proposed a reasonable early stage biosynthetic route for demethoxyviridin (Fig. 3).

In the present study, we have demonstrated that five enzymes, VidE, VidG, VidR, VidN, and VidO, are involved in the early stage biosynthesis of demethoxyviridin by their co-expression in *A. oryzae* NSAR1 (Fig. 4). We obtained three stable biosynthetic intermediates, **13**, **14**, and **15**, from the five-gene co-expression strain without adding any precursors, thus laying the foundation for the reconstitution of demethoxyviridin biosynthesis in *A. oryzae*, as we previously achieved for helvolic acid[42].

The cleavage of the pregnane side chain of progesterone in fungi has been investigated for many years, but the biochemical basis for this process remained unclear. Here, we demonstrated that VidF, VidP, and VidH are able to cleavage progesterone to the corresponding 17-keto steroids, suggesting that the pregnane side-chain cleavage mediated by VidF, VidP, and VidH is the predominant mechanism in fungal steroid biosynthesis (Fig. 7). Consistent with this speculation, a homology search of VidF,

VidP, and VidH using MultiGeneBlast[43] at the NCBI database revealed that these gene clusters are widely distributed in fungal genomes, suggesting that the three-enzyme-mediated pregnane side-chain cleavage is likely to be a conserved sterol metabolic pathway in the fungal kingdom (Supplementary Fig. 10).

This study generated 14 intermediates, including 8 new compounds, which provided fresh insights into the structure–activity relationship of demethoxyviridin for PI3K inhibition. We found that the β-OH at C-1 is important for inhibiting PI3K, as **1** and **2** exhibited more potent activities than their corresponding 1-dehydroxyl products **3** and **4**, respectively. The C1β-hydroxylation by the CYP enzyme VidK is rare, as enzymes capable of specifically catalyzing C1β-hydroxylation have not been reported, although an enzyme catalyzing the dual hydroxylations of C6 and C1 has been reported[44], thus indicating that VidK will be a tool for the C1β-hydroxylation of steroidal compounds. We also observed that the aromatic ring C is important for the activity (Fig. 8a). The aromatization of ring A in estrogen biosynthesis has been well described, but ring C aromatization has seldom been reported in natural products. Thus, VidA and VidD, which are involved in the aromatization of ring C, should be useful for steroid transformation.

In conclusion, we identify the biosynthetic gene cluster and establish the complete biosynthetic pathway for demethoxyviridin, one of the most representative furanosteroids with potent inhibitory activity against PI3K. In addition, we have obtained 14 demethoxyviridin intermediates, including eight previously unknown compounds, via targeted gene deletion, thus providing

the insights into the structure–activity relationships of furanosteroids. These findings provide important knowledge toward the biosyntheses of furanosteroids and explorations of combinatorial biosyntheses to expand the chemical diversity of furanosteroids.

## Methods

**General experimental procedures**. Methanol (MeOH) was purchased from Yuwang Industrial Co., Ltd. (Yucheng, China). Acetonitrile (MeCN) was obtained from Oceanpak Alexative Chemical Co., Ltd. (Gothenburg, Sweden). Ethyl acetate (EtOAc) and chloroform (CHCl$_3$) were analytical grade from Fine Chemical Co., Ltd. (Tianjin, China). The compound standards (**20**, **22**, and **23**) were purchased from J&K Scientific Co., Ltd. (Beijing, China).

Primer synthesis and DNA sequencing were performed by Sangon Biotech Co., Ltd. (Shanghai, China). PCR was performed using KOD FX DNA polymerase, KOD Plus DNA polymerase (TOYOBO, Osaka, Japan), and Taq DNA polymerase (TaKaRa, Dalian, China). T4 DNA polymerase and an In-Fusion® HD Cloning Kit were used, as recommended by the manufacturer (TaKaRa, Dalian, China). DNA restriction enzymes and other DNA modification reagents were purchased from Thermo Fisher Scientific (Shenzhen, China) and TOYOBO Co., Ltd. (Osaka, Japan).

UV data were obtained with a JASCO V-550 UV/vis spectrometer (Jasco International Co., Ltd., Tokyo, Japan). IR data were acquired with a JASCO FT/IR-480 plus spectrometer (Jasco International Co., Ltd., Tokyo, Japan). Optical rotations were determined on a JASCO P1020 digital polarimeter (Jasco International Co., Ltd., Tokyo, Japan). ECD spectra were recorded on a JASCO J-810 spectrophotometer (Jasco International Co., Ltd., Tokyo, Japan), using MeOH as the solvent. The melting points were measured on an X-5 micromelting point apparatus (Beijing TECH Instrument Co., Ltd., Beijing, China) without corrections. The ESIMS spectra were recorded using a Bruker amaZon SL mass spectrometer (Bruker Daltonics Int., Boston, MA, USA) and the HRESIMS spectra were performed obtained with a Waters Synapt G2 mass spectrometer (Waters Corporation, Milford, MA, USA). The 1D and 2D NMR spectra were recorded on Bruker AV 300, Bruker AV 400, and Bruker AV 600 spectrometers (Bruker BioSpin Group, Faellanden, Switzerland), using the following solvent signals (DMSO-$d_6$: $\delta_H$ 2.50/$\delta_C$ 39.5; CD$_3$OD: $\delta_H$ 3.30/$\delta_C$ 49.0; CDCl$_3$: $\delta_H$ 7.26/$\delta_C$ 77.0) as internal standards.

Column chromatography (CC) was performed on ODS resin (50 μm, YMC Co., Ltd., Tokyo, Japan). Analytical HPLC was conducted with a Dionex HPLC system equipped with an Ultimate 3000 pump, an Ultimate 3000 DAD, an Ultimate 3000 column compartment, an Ultimate 3000 autosampler (Thermo Fisher Scientific Inc., Sunnyvale, USA), and an Alltech (Grace) 2000ES evaporative light scattering detector (ELSD) (Alltech Co., Ltd., Portland, USA), using a Phenomenex Gemini C$_{18}$ column (4.6 × 250 mm, 5 μm) (Phenomenex Inc., Los Angeles, CA, USA). The preparative HPLC was performed on a Shimadzu LC-6-AD liquid chromatography system (Shimadzu Inc., Kyoto, Japan) with an SPD-20A detector, using a YMC-Pack ODS-A column (10.0 × 250 mm, 5 μm) (YMC Inc., Kyoto, Japan). Medium pressure liquid chromatography (MPLC) was performed with a UV preparative detector, a dual pump gradient system, and a Dr. Flash II fraction collector system (Lisui E-Tech Co., Ltd., Shanghai, China).

**Strains and media**. *Nodulisporium* sp. (no. 65-12-7-1), a high producer of demethoxyviridin, has been deposited in GenBank as specimen KC894854. The strain was maintained on potato dextrose agar (PDA), and cultivated at 28 °C, 180 rpm in maltose medium (3% maltose, 0.25% malt extract, 0.15% yeast extract, 0.2% KH$_2$PO$_4$, 0.1% MgSO$_4$, 0.4% CaCO$_3$) for 3 days, and was used as the source for whole-genome sequencing. The total RNA for RT-PCR and RNA-Seq was extracted from *Nodulisporium* sp. (no. 65-12-7-1), grown on maltose medium (viridins producing) and Czapek medium (viridins non-producing, 3% sucrose, 0.3% NaNO$_3$, 0.05% KCl, 0.1% K$_2$HPO$_4$, 0.05% MgSO$_4$–7H$_2$O), at 28 °C, 180 rpm, for 2 days.

For the production of **1** and other metabolites, the Cas9-expressing strain (JN1001)[30] and gene deletion strains were grown in 10 mL maltose medium at 28 °C, 180 rpm, for 2 days. The cells were then transferred into 500 mL Erlenmeyer flasks containing 100 mL maltose medium and the culture was continued at 28 °C on a rotary shaker at 180 rpm, for 2 days.

*Aspergillus oryzae* NSAR1 (*niaD⁻*, *sC⁻*, *ΔargB*, and *adeA⁻*)[34] was used as the host for heterologous gene expression. The *A. oryzae* NSAR1 transformants were cultured in 10 mL DPY medium (2% dextrin, 1% polypeptone, 0.5% yeast extract, 0.5% KH$_2$PO$_4$, 0.05% MgSO$_4$–7H$_2$O) at 28 °C, 180 rpm for 2 days. The cells were then transferred into Czapek-Dox (CD) medium (2% starch, 1% polypeptone, 0.3% NaNO$_3$, 0.2% KCl, 0.1% K$_2$HPO$_4$, 0.05% MgSO$_4$–7H$_2$O, 0.002% FeSO$_4$–7H$_2$O, pH 5.5) to induce the expression of exogenous genes under the *amyB* promoter, and the cultures were shaken for 5 more days.

*E. coli* DH5α was used for cloning, while *E. coli* BL21-Codon Plus (DE3) was used for expression. These strains were grown in Luria−Bertani (LB) medium supplemented with appropriate antibiotics.

**Whole-genome sequencing and analysis**. The whole-genome sequencing of *Nodulisporium* sp. (no. 65-12-7-1) was performed by Sangon Biotech Co., Ltd. (Shanghai, China) with an Illumina HiSeq 2500 system. The SOAPdenovo version 2.04 (http://soap.genomics.org.cn/soapdenovo.html) sequence assembly system was used to produce 381 contigs, covering ~36.9 Mb. Gene prediction was then performed with AUGUSTUS (http://bioinf.uni-greifswald.de/webaugustus/), and manually revised according to the coding sequences obtained from the transcriptome and the homologous genes found in the NCBI database.

**RNA preparation and reverse transcription PCR (RT-PCR)**. Mycelia were collected from the maltose medium and the Czapek medium as described above, and ground in liquid nitrogen with a mortar. RNA was then isolated using an RNeasy Plant Mini Kit (QIAGEN), according to the manufacturer's protocol. To eliminate the genomic DNA, RNA samples were treated with RNase-free DNase I (TaKaRa). The first strand cDNA was synthesized with the PrimeScript™ II 1st Strand cDNA Synthesis Kit (TaKaRa). In order to suppress the interference from the genomic DNA, all of the primers used for RT-PCR flanked an intron, and are listed in Supplementary Table 1, so that a smaller amplicon should be generated from the total RNA, as compared to that from the genomic DNA (Supplementary Figs. 3 and 4). Glyceraldehyde-3-phosphate dehydrogenase (GAPDH) was used as a reference.

**Construction of gene inactivation mutants**. The gene disruption in JN1001 was performed by the CRISPR/Cas9 system established by our laboratory[30]. The genes (*g3262*, *g3264*, *g3265*, and *g3285*) and the *vid* genes of the cluster, except for two transporter genes (*vidB* and *vidL*) and two Bet v1-like genes (*vidC* and *vidI*), were deleted.

For the preparation of the in vitro transcriptional gRNA, the gRNA cassettes containing the T7 promoter, the protospacer sequence, and the synthetic gRNA scaffold for targeting genes were PCR amplified from the plasmid pUCm-gRNAscaffold-*eGFP*, using the primers listed in Supplementary Table 2, and inserted into pUCm-T to generate the corresponding plasmids shown in Supplementary Table 3. These vectors were used as templates for PCR amplification by the primers pUCm-F/gRNA-R, and the resulting PCR products were used for the in vitro transcription of gRNAs with the T7 RiboMAX™ Express Large Scale RNA Production System (Promega, China).

The linear *neo* marker gene cassette was amplified from the plasmid pBSKII-PtrPC-*neo*-TtrPC, using the primers PtrpC-XbaI-F/TtrpC-HindIII-R listed in Supplementary Table 2.

For transformation of JN1001, the strain was inoculated into 10 mL DPY medium and cultivated at 28 °C, 180 rpm, for 2 days. The cell culture was then transferred into 100 mL DPY medium and cultivated for 24 h. The mycelia were collected and digested using 1% Yatalase (Takara) in 0.6 M (NH$_4$)$_2$SO$_4$, 50 mM maleic acid, pH 5.5 at 30 °C for 3 h. The resulting protoplasts were then separated from mycelia by filtration and washed with TF solution 2 (1.2 M sorbitol, 50 mM CaCl$_2$–2H$_2$O, 35 mM NaCl, 10 mM Tris-HCl, pH 7.5), and diluted to a concentration of 2 × 10$^7$ cells mL$^{-1}$. Then, the linear neo marker gene cassette and the in vitro transcriptional gRNA were added to the 200 μL protoplasts solution, and incubated on ice for 30 min, which was blended with 1.35 mL solution 3 (60% PEG4000, 50 mM CaCl$_2$–2H$_2$O, 10 mM Tris-HCl, pH 7.5) and incubated at the room temperature for 20 min. The resulting solution was then diluted with 10 mL solution 2 and centrifuged at 420 × *g* for 10 min. The precipitates were suspended with 200 μL solution 2 and added on the M medium (0.2% NH$_4$Cl, 0.1% (NH$_4$)$_2$SO$_4$, 0.05% KCl, 0.05% NaCl, 0.1% KH$_2$PO$_4$, 0.05% MgSO$_4$–7H$_2$O, 0.002% FeSO$_4$–7H$_2$O, 2% glucose and 1.2 M sorbitol as well as 200 μg mL$^{-1}$ G418, pH 5.5) with 1.5% agar, and covered with the upper M medium containing 0.8% agar. The plates were incubated at 30 °C for 5–7 days. All of the mutants used in this report are listed in Supplementary Table 4.

**Expression and purification of VidF**. The full-length *vidF* gene was amplified from the *Nodulisporium* sp. (no. 65-12-7-1) cDNA, with the primers listed in Supplementary Table 2, and ligated into the pET-28a (+) vector linearized by *Nde*I and *Not*I cleavage to yield the *E. coli* expression vector pET-28a (+)-*vidF*. For the expression of the VidF protein, *E. coli* BL21-Codon Plus (DE3) was transformed with the pET-28a (+)-*vidF* plasmid. The transformant was incubated in LB medium supplemented with 50 mg L$^{-1}$ kanamycin sulfate, at 37 °C/200 rpm for 16 h. Gene expression was induced by the addition of 0.1 mM IPTG when the OD$_{600}$ reached 0.6, followed by further incubation with shaking at 200 rpm at 18 °C for 16 h. The cells were harvested by centrifugation, resuspended in lysis buffer (50 mM Tris-HCl, 200 mM NaCl, 5 mM imidazole, 5% glycerol, pH 8.0), and lysed on ice by sonication. After centrifugation, the supernatant was applied to a Ni-NTA affinity column, which was washed with 30 column volumes of wash buffer (50 mM Tris-HCl, 150 mM NaCl, 20 mM imidazole, 5% glycerol, pH 8.0). The His-tagged protein was eluted with 5 column volumes of elution buffer (50 mM Tris-HCl, 200 mM NaCl, 300 mM imidazole, 5% glycerol, pH 8.0). The purified enzyme was analyzed by sodium dodecyl sulfate polyacrylamide gel electrophoresis (SDS-PAGE) (Supplementary Fig. 8). A Bradford Protein Assay (Bio-Rad) was used to calculate protein concentration.

**Enzyme assay with purified protein**. The enzymatic reaction of VidF with 3-dihydrovirone (**11**) was performed in a 100 μL reaction mixture, containing 50 mM Tris-HCl (pH 8.0), 125 μM NADPH, 0.25 mM of **11**, 1 mM FAD, and 2 μM of VidF. The reaction was performed at 25 °C for 24 h, with shaking at 150 rpm. The mixture was then extracted with EtOAc twice. After evaporation, the extract was analyzed by HPLC.

**Preparation of fungal expression plasmids**. To construct the fungal expression vectors for each gene (*vidE*, *vidG*, *vidN* *vidO*, *vidR*, *vidF*, *vidH*, and *vidP*), each gene was amplified from *Nodulisporium* sp. (no. 65-12-7-1) genomic DNA, with the primers listed in Supplementary Table 2, purified, and ligated into the pTAex3 or pUSA vector, using an In-Fusion HD Cloning Kit according to the manufacturer's protocol, to yield the corresponding plasmids.

For the construction of the plasmid harboring three genes (*vidE*, *vidN*, and *vidO*) or the plasmid harboring two genes (*vidH* and *vidP*), DNA fragments including the *amyB* promoter and terminator were amplified from pTAex3-*vidE*, pTAex3-*vidN*, pTAex3-*vidO*, pTAex3-*vidH*, and pTAex3-*vidP*, with the primers listed in Supplementary Table 2, and then the PCR products were inserted into the *Xba*I-digested pAdeA vector, using the In-Fusion® HD Cloning Kit, to construct the expression plasmids listed in Supplementary Table 3.

**Transformation of *A. oryzae* NSAR1**. Transformation of *A. oryzae* NSAR1 was performed by the similar protoplast–polyethylene glycol method as described above in the transformation of JN1001 strain. To coexpress *vidE*, *vidG*, *vidN*, *vidO*, and *vidR*, the plasmids, pTAex3-*vidG*, pUSA-*vidR*, and pAdeA-*vidE*-*vidN*-*vidO*, were used for the transformation to generate the strain JA1. To express *vidP* and *vidH*, respectively, the plasmids, pTAex3-*vidP* and pTAex3-*vidH*, were used for the transformation to generate the strains JA2 and JA3, respectively. For the construction of strain JA4 harboring three genes (*vidF*, *vidH*, and *vidP*), the plasmid pTAex3-*vidF*, together with pAdeA-*vidH*-*vidP*, was used for the transformation. All of the transformants used in this work are listed in Supplementary Table 4.

**Feeding experiments**. The *A. oryzae* NSAR1 transformant strain (JA2, JA3, or JA4) was inoculated into 10 mL DPY medium and cultured for 2 days, and then fermentation was performed in 50 mL CD medium for the expression of exogenous genes. After cultivation for 1 day, 1.0 mg of substrate, dissolved in 20 μL DMSO, was added to the culture, which was further incubated for 4 days. The culture was extracted with an equal volume of EtOAc and evaporated to dryness. The dried extract was dissolved in MeOH for the HPLC analysis.

**HPLC analysis and isolation of each metabolite**. Products from the mutants, the transformants, and the in vitro reaction mixtures were analyzed by HPLC, eluted with a MeOH−H₂O system (35:75 for 10 min, a linear gradient from 35:80 to 100:0 within the following 38 min, and 100:0 for 10 additional min), at a flow rate of 1 mL min⁻¹.

For compound isolation, 1−3 L of the culture media were extracted with EtOAc thrice, and then the crude extract was subjected to MPLC by ODS column chromatography, eluted with a gradient of MeOH–H₂O, or subjected to silica-gel column chromatography using a CHCl₃–MeOH gradient, and further purification by preparative HPLC to afford the main metabolites. The detailed purification procedures for each compound are described in the Supplementary Methods.

**In vitro PI3K gamma kinase assay**. The in vitro PI3K gamma kinase assay was carried out using a PI3-Kinase (human) HTRF™ Assay kit (Millipore, USA) according to the manufacturer's protocol. Briefly, the PI3K gamma kinase was incubated with 10 μM sample (tested compounds) in the reaction buffer containing 10 μM PIP2 in the wells of a 384-well opaque black plate (PerkinElmer, USA) at room temperature. Kinase reactions were initiated by adding 10 μM ATP and incubating the plate at room temperature for 1 h. The reactions were quenched by adding 5 μL of Stop Solution (containing EDTA and biotin-PIP3), and then the detection buffer (containing the Europium-labeled anti-GST antibody, the GST-tagged receptor of the phosphoinositide (GRP1) PH domain, and streptavidin–APC) was added to each well. Following 2 h incubation in the dark, the HTRF signal was measured with a 2104 EnVision® Multilabel Reader (PerkinElmer, USA) in the time-resolved fluorescence mode, set for excitation at 320 nm and dual emission detection at 615 nm (Eu) and 665 nm (APC). The emission ratio (ER) of 665 nm emission signal to 615 nm emission signal was then calculated and the inhibition ratio of a certain sample on PI3K was calculated according to the following formula: PI3K inhibition (%) = (ER$_{sample}$ − ER$_{0\%}$)/(ER$_{100\%}$ − ER$_{0\%}$ × 100%. ER$_{0\%}$ means 0% inhibition, which was calculated from the reaction of PI3K, PIP2, and ATP in the absence of sample. ER$_{100\%}$ means 100% inhibition, which was calculated from the reaction of PIP2 and ATP without PI3K. BEZ235 was used as a positive control. For active inhibitors (relative inhibitory activity ≥50%), the IC$_{50}$ values were estimated using the "Sigmoidal dose-response (variable slope)" equation in the GraphPad Prism® Version 5.0 software, with the maximum and minimum of the curve constrained to 100 and 0, respectively.

**Cytotoxicity assay**. The human breast cancer MCF-7 cell line (ATCC HTB-22) was used in the cytotoxicity assay. The cell line was cultured in DMEM (Gibco, Life Technologies, USA), supplemented with 10% fetal bovine serum (Gibco, Life Technologies, USA), in a 5% CO₂ atmosphere at 37 °C. The cytotoxicity assay was performed using the CCK-8 (Dojindo Laboratories, Kumamoto, Japan) method in 96-well microplates, with cisplatin (DDP) (Sigma, USA) as the positive control. After a treatment with 40 μM of each compound (Dissolved with DMSO) for 48 h, the CCK-8 solution (10 μL) was added to each well and incubated for 2 h at 37 °C. The absorbance (Ab) of samples was detected at 450 nm using a microtiter plate reader (TriStar LB941, Berthold Technologies, Germany). The cell viability of each sample was calculated by the following formula: [(Ab$_{sample}$ − Ab$_{blank}$)/(Ab$_{DMSO}$ − Ab$_{blank}$)] × 100%, and the IC$_{50}$ values were calculated using the "Sigmoidal dose-response (variable slope)" equation in the GraphPad Prism® Version 5.0 software.

**Compound spectral data**. Compound **1**: white amorphous powder; HRESIMS (positive) *m/z* 323.0923 [M+H]$^+$ (calcd. for C₁₉H₁₅O₅, 323.0919), see Supplementary Fig. 14; NMR spectra see Supplementary Fig. 15. The NMR data are in good agreement with those of demethoxyviridin[26] (Supplementary Note 1).

Compound **2**: white amorphous powder; HRESIMS (positive) *m/z* 325.1071 [M+H]$^+$ (calcd. for C₁₉H₁₇O₅, 325.1076), see Supplementary Fig. 16; NMR spectra see Supplementary Figs. 17–19; NMR data see Supplementary Table 6. The NMR data are in good agreement with those of demethoxyviridiol[45] (Supplementary Note 2).

Compound **3**: white amorphous powder; HRESIMS (positive) *m/z* 307.0966 [M+H]$^+$ (calcd. for C₁₉H₁₅O₄, 307.0970), see Supplementary Fig. 20; NMR spectra see Supplementary Figs. 21–23; NMR data see Supplementary Table 7. The NMR data are in good agreement with those of dehydroxydemethoxyviridin[46] (Supplementary Note 3).

Compound **4**: white amorphous powder; HRESIMS (positive) *m/z* 309.1129 [M+H]$^+$ (calcd. for C₁₉H₁₇O₄, 309.1127), see Supplementary Fig. 24; UV (MeOH) λ$_{max}$ (log ε) 204 (4.36), 248 (4.56), 316 (4.26) nm; ECD (*c* 1.6 × 10⁻⁴ M, MeOH) λ$_{max}$ (Δε): 215 (+10.47), 225 (+11.43), 258 (−7.49), 302 (+3.17), 329 (+1.75), 354 (−4.56); IR (KBr) ν$_{max}$ 3514, 2947, 2866, 1689, 1662, 1585, 1439, 1057, 968 cm⁻¹; NMR spectra, see Supplementary Figs. 25–27; NMR data see Supplementary Table 8; and **4** was identified as (3*S*,11b*R*)-3-hydroxy-11b-methyl-1,2,3,7,8,11b-hexahydrocyclopenta[7,8]phenanthro[10,1-*bc*]furan-6,9-dione (Supplementary Note 4).

Compound **5**: colorless plate-like crystals; m.p. 236.4–238.7 °C; HRESIMS (positive) *m/z* 325.1446 [M+H]$^+$ (calcd. for C₂₀H₂₁O₄, 325.1440), see Supplementary Fig. 28; UV (MeOH) λ$_{max}$ (log ε) 205 (3.81), 221 (3.66), 296 (3.59), 342 (3.68) nm; ECD (*c* 3.1 × 10⁻⁴ M, MeOH) λ$_{max}$ (Δε): 219 (+0.85), 240 (−10.92), 287 (−5.23), 340 (+7.67); IR (KBr) ν$_{max}$ 3410, 2962, 2877, 1720, 1651, 1616, 1454, 1377, 1053 cm⁻¹; NMR spectra see Supplementary Figs. 29–31; NMR data see Supplementary Table 9; X-ray crystallographic data see Supplementary Note 18; and **5** was identified as (3*S*,6b*R*,9a*S*,11b*R*)-3-hydroxy-9a,11b-dimethyl-1,2,3,6b,7,8,9a,11b-octahydrocyclopenta[7,8]phenanthro[10,1-*bc*]furan-6,9-dione (Supplementary Note 5).

Compound **6**: yellowish amorphous powder; HRESIMS (positive) *m/z* 327.1593 [M+H]$^+$ (calcd. for C₂₀H₂₃O₄, 327.1596), see Supplementary Fig. 32; UV (MeOH) λ$_{max}$ (log ε) 205 (3.75), 270 (3.67), 312 (3.45) nm; ECD (*c* 3.1 × 10⁻⁴ M, MeOH) λ$_{max}$ (Δε): 209 (+1.66), 225 (−7.00), 244 (+1.23), 267 (−4.46), 302 (+4.93), 345 (−1.47); IR (KBr) ν$_{max}$ 3487, 2947, 2873, 1728, 1658, 1616, 1377, 1041 cm⁻¹; NMR spectra see Supplementary Figs. 33–35; NMR data see Supplementary Table 10; and **6** was identified as (3*S*,6b*R*,9a*S*,11b*R*)-3-hydroxy-9a,11b-dimethyl-1,2,3,6b,7,8,9a,10,11,11b-decahydrocyclopenta[7,8]ph-enanthro[10,1-*bc*]furan-6,9-dione (Supplementary Note 8).

Compound **7**: white amorphous powder; HRESIMS (positive) *m/z* 327.1602 [M+H]$^+$ (calcd. for C₂₀H₂₃O₄, 327.1596), see Supplementary Fig. 36; UV (MeOH) λ$_{max}$ (log ε) 206 (4.05), 229 (3.84), 264 (3.77), 300 (3.63) nm; ECD (*c* 3.7 × 10⁻⁴ M, MeOH) λ$_{max}$ (Δε): 219 (−7.77), 263 (−4.32), 321 (+2.28); IR (KBr) ν$_{max}$ 3433, 2939, 2877, 1693, 1658, 1531, 1354, 1103 cm⁻¹; NMR spectra see Supplementary Figs. 37–39; NMR data see Supplementary Table 11; and **7** was identified as (6b*R*,9*S*,9a*S*,11b*R*)-9-hydroxy-9a,11b-dimethyl-1,2,6b,7,8,9,9a,10,11,11b-decahydrocyclopenta[7,8]phenanthro[10,1-*bc*]furan-3,6-dione (Supplementary Note 7).

Compound **8**: colorless plate-like crystals; m.p. 187.1–189.6 °C; HRESIMS (positive) *m/z* 329.1753 [M+H]$^+$ (calcd. for C₂₀H₂₅O₄, 329.1753), see Supplementary Fig. 40; UV (MeOH) λ$_{max}$ (log ε) 205 (3.87), 263 (3.75), 310 (3.62) nm; ECD (*c* 3.0 × 10⁻⁴ M, MeOH) λ$_{max}$ (Δε): 223 (−7.73), 244 (+1.28), 267 (−5.22), 306 (+2.17); IR (KBr) ν$_{max}$ 3406, 2947, 2873, 1651, 1616, 1454, 1346, 1053 cm⁻¹; NMR spectra see Supplementary Figs. 41–43; NMR data see Supplementary Table 12; X-ray crystallographic data see Supplementary Note 19; and **8** was identified as (3*S*,6b*R*,9*S*,9a*S*,11b*R*)-3,9-dihydroxy-9a,11b-dimethyl-2,3,6b,7,8,9,9a,10,11,11b-decahydrocyclopenta[7,8]phenanthro[10,1-*bc*]furan-6 (1*H*)-one (Supplementary Note 6).

Compound **9**: white amorphous powder; HRESIMS (positive) *m/z* 371.1865 [M+H]$^+$ (calcd. for C₂₂H₂₇O₅, 371.1858), see Supplementary Fig. 44; UV (MeOH) λ$_{max}$ (log ε) 205 (3.73), 218 (3.55), 264 (3.70), 310 (3.60) nm; ECD (*c* 3.2 × 10⁻⁴ M, MeOH) λ$_{max}$ (Δε): 223 (−7.02), 242 (+2.03), 267 (−5.83), 306 (+1.61), 347 (−0.59); IR (KBr) ν$_{max}$ 3487, 3425, 2935, 2870, 1716, 1658, 1620, 1377, 1277, 1057 cm⁻¹;

NMR spectra see Supplementary Figs. 45−47; NMR data see Supplementary Table 13; and 9 was identified as (3S,6bR,9S,9aS,11bR)-3-hydroxy-9a,11b-dimethyl-6-oxo-1,2,3,6,6b,7,8,9,9a,10,11,11b-dodecahydrocyclopenta[7,8]phenanthro[10,1-*bc*]furan-9-yl acetate (Supplementary Note 9).

Compound 10: yellowish amorphous powder; HRESIMS (positive) *m/z* 353.1758 [M+H]$^+$ (calcd. for $C_{22}H_{25}O_4$, 353.1753), see Supplementary Fig. 48; NMR spectra see Supplementary Figs. 49−51; NMR data see Supplementary Table 14. The NMR data are in good agreement with those of virone[46] (Supplementary Note 11).

Compound 11: yellowish amorphous powder; HRESIMS (positive) *m/z* 355.1915 [M+H]$^+$ (calcd. for $C_{22}H_{27}O_4$, 355.1909), see Supplementary Fig. 52; NMR spectra see Supplementary Fig. 53. The NMR data are in good agreement with those of 3-dihydrovirone[47] (Supplementary Note 10).

Compound 12: white amorphous powder; HRESIMS (positive) *m/z* 363.2545 [M+H]$^+$ (calcd. for $C_{22}H_{35}O_4$, 363.2535), see Supplementary Fig. 54; NMR spectra see Supplementary Fig. 55. The NMR data are in good agreement with those of nodulisporisteroid C[41] (Supplementary Note 12).

Compound 13: colorless plate-like crystals; m.p. 218.2–221.7 °C; HRESIMS (positive) *m/z* 373.2019 [M+H]$^+$ (calcd. for $C_{22}H_{29}O_5$, 373.2015), see Supplementary Fig. 56; UV (MeOH) $\lambda_{max}$ (log $\varepsilon$) 204 (3.90), 216 (3.86), 255 (3.96) nm; ECD (*c* 3.2 × 10$^{-4}$ M, MeOH) $\lambda_{max}$ ($\Delta\varepsilon$): 218 (−5.16), 238 (+3.36), 287 (+3.69), 356 (−0.43); IR (KBr) $\nu_{max}$ 3375, 3197, 2943, 2870, 1693, 1577, 1454, 1354, 1057, 987 cm$^{-1}$; NMR spectra see Supplementary Figs. 57−59; NMR data see Supplementary Table 15; X-ray crystallographic data see Supplementary Note 20; and 13 was identified as (3S,5aR,6bR,9S,9aS,11bR)-9-acetyl-3,5a-dihydroxy-9a,11b-dimethyl-2,3,4,5a,6b,7,8,9,9a,10,11,11b-dodecahydrocyclopenta[7,8]phenanthro [10,1-*bc*]furan-6(1*H*)-one (Supplementary Note 13).

Compound 14: yellowish amorphous powder; HRESIMS (positive) *m/z* 397.1999 [M+Na]$^+$ (calcd. for $C_{22}H_{30}O_5$ Na, 397.1991), see Supplementary Fig. 60; UV (MeOH) $\lambda_{max}$ (log $\varepsilon$) 207 (4.02) nm; ECD (*c* 1.6 × 10$^{-4}$ M, MeOH) $\lambda_{max}$ ($\Delta\varepsilon$): 220 (+2.96), 286 (+7.11); IR (KBr) $\nu_{max}$ 3529, 3356, 3232, 2935, 2858, 1689, 1450, 1362, 1065, 991 cm$^{-1}$; NMR spectra see Supplementary Figs. 61−63; NMR data see Supplementary Table 16; and 14 was identified as 1-((3S,5aR,6S,6bR,9S,9aS,11bR)-3,5a,6-trihydroxy-9a,11b-dimethyl-1,2,3,4,5a,6,6b,7,8,9,9a,10,11,11b-tetradecahydrocyclopenta[7,8]phenanthro[10,1-*bc*]furan-9-yl)ethan-1-one (Supplementary Note 14).

Compound 15: yellowish amorphous powder; HRESIMS (positive) *m/z* 395.1821 [M+Na]$^+$ (calcd. for $C_{22}H_{30}O_5$ Na, 395.1834), see Supplementary Fig. 64; UV (MeOH) $\lambda_{max}$ (log $\varepsilon$) 207 (4.10), 247 (3.96) nm; ECD (*c* 3.4 × 10$^{-4}$ M, MeOH) $\lambda_{max}$ ($\Delta\varepsilon$): 205 (+17.86), 225 (−7.30), 258 (+6.89), 279 (+5.46), 325 (−1.16); IR (KBr) $\nu_{max}$ 3515, 3315, 2969, 2931, 2882, 2839, 1677, 1446, 1383, 1365, 1232, 1212, 1093, 1050, 998 cm$^{-1}$; NMR spectra see Supplementary Figs. 65−67; NMR data see Supplementary Table 17; and 15 was identified as (5aR,6S,6bR,9S,9aS,11bR)-9-acetyl-5a,6-dihydroxy-9a,11b-dimethyl-1,4,5a,6,6b,7,8,9,9a,10,11,11b-dodecahydrocyclopenta[7,8]phenanthro[10,1-bc]furan-3(2*H*)-one (Supplementary Note 15).

Compound 21: white amorphous powder; HRESIMS (positive) *m/z* 331.2281 [M+H]$^+$ (calcd. for $C_{21}H_{31}O_3$, 331.2273), see Supplementary Fig. 68; NMR spectra see Supplementary Fig. 69. The NMR data are in good agreement with those of testosterone acetate[48] (Supplementary Note 16).

Compound 24: white amorphous powder; HRESIMS (positive) *m/z* 303.1976 [M+H]$^+$ (calcd. for $C_{19}H_{27}O_3$, 303.1960), see Supplementary Fig. 70; NMR spectra see Supplementary Fig. 71. The NMR data are in good agreement with those of testolactone[49] (Supplementary Note 17).

**Data availability**. The sequence data for the *vid* gene have been deposited at GenBank under accession MG886384. The crystallographic data of small-molecule compounds related to this manuscript have been deposited at the Cambridge Crystallographic Data Center: CCDC 1821761−1821763. All relevant data are available from the corresponding authors on request.

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

## Acknowledgements

We thank Professor K. Gomi (Tohoku University) and Professor K. Kitamoto (The University of Tokyo) for the *Aspergillus oryzae* NSAR1 heterologous expression system. This work was mainly supported by grants from the National Natural Science Foundation of China (31670036, 3171101305, and 81673315), the 111 Project of Ministry of Education of the People's Republic of China (B13038), the JST/NSFC Strategic International Collaborative Research Program, Japanese-Chinese Collaborative Research Program, Chang Jiang Scholars Program (Hao Gao, 2017) from the Ministry of Education of China, Guangdong Special Support Program (2016TX03R280), Guangdong Province Universities and Colleges Pearl River Scholar Funded Scheme (Hao Gao, 2014), and K. C. Wong Education Foundation (Hao Gao, 2016), Guangzhou Science and Technology Project (201707010266), Kobayashi International Scholarship Foundation, a Grant-in-Aid for Scientific Research from the Ministry of Education, Culture, Sports, Science and Technology, Japan (JSPS KAKENHI grant numbers JP15H01836 and JP16H06443).

## Author contributions

D.H., I.A., and H.G. designed the research. G.-Q.W., G.-D.C., and S.-Y.Q. performed the experiments. G.-Q.W., G.-D.C., S.-Y.Q., D.H., T.A., S.-Y.L., J.-M.L., C.-X.W., X.-S.Y., I. A., and H.G. analyzed the data, and wrote the paper.
