## [Peer Review File · Nature Communications]

Reviewers' comments:

Reviewer #1 (Remarks to the Author):

The manuscript entitled "Biosynthetic Pathway for Furanosteroid Antibiotic Demethoxyviridin and Identification of an unusual Pregnane Side-chain Cleavage" by Wang et al describes the sequencing of the demethoxyviridin biosynthetic gene cluster from the fungus *Nodulisporium* sp No 65-71-2-1, the confirmation that the gene cluster encodes the natural products via loss of function mutagenesis, dissection of the late stages of the biosynthetic pathway via systematic deletion of individual biosynthetic genes and characterization of the newly formed intermediates, heterologous expression of the early stages of the gene cluster in *Aspergillus oryzae* NSAR1 confirming the genes responsible for the early stage of the biosynthesis, enzymatic assay with recombinant purified BVFMO unambiguously demonstrating that this enzyme is responsible for side-chain cleavage, and heterologous expression of the side-chain cleavage three gene cluster in *A. oryzae* demonstrating its substrate tolerance.

This work clearly identifies the biosynthetic pathway for formation of demethoxyviridin and by extension, likely many of the related furanosteroids. While these compounds have been studied extensively, identification of the biosynthetic gene cluster has eluded researchers. Wang and coworkers used a new approach for identification of the gene cluster. Typically researchers mine genomes for isoprenoid pathways by looking for a core terpene cyclase gene. Because furanosteroids are generated from lanosterol, which is produced via the main primary steroid biosynthetic pathway and not a terpene cyclase, this approach could not identify the pathway. Wang and coworkers instead mined the genome for clustered P450, which are required for the significant oxidative processing of the compound. This enabled them to rapidly find putative gene clusters, evaluate them based on transcription and unambiguously identify the pathway via loss of function mutagenesis. This is an important addition to the strategies used to identify gene clusters in fungi.

The authors clearly deduce the order of the steps in the biosynthesis and the role of many of the genes responsible for demethoxyviridin production. This work will have a significant impact on the field. They show that C ring aromatization occurs similarly to A ring aromatization in estradiol biosynthesis and that loss of the pregnane side-chain occurs in this scaffold occurs via a BVFMO as is known for loss of the sidechain from progesterone in fungi.

Lastly the authors isolated and convincingly characterize 14 intermediates, 8 of which are new compounds. Using these compounds they are able to add to the structure activity relationship for the viridian class of PI3K inhibitors.

This work will impact researchers far beyond the biosynthesis community. This manuscript should be published in Nature Communications following minor revisions.

Minor revision:

1. Line 151. C10 demethylation, rather than C14, is more appropriate comparison since it likewise generates an aromatic ring. Please adjust.
2. Lines 239-242. The authors indicate that this experiment demonstrates a "conserved sterol metabolic pathway". It does not. By demonstrating that the viridian biosynthetic proteins can also process progesterone, the authors have demonstrated that the enzymes have some degree of substrate tolerance. To determine if this is a conserved pathway, that is the pathway is present in many and diverse fungal genomes, the authors would need to mine multiple genomes and confirm that *vidF*, *vidH*, and *vidP* orthologs are present. Please correct.

Reviewer #2 (Remarks to the Author):

This is excellent work by Abe, Gao, and co-workers. An elegant screening was performed to find the first gene cluster of furanosateroids. Gene inactivation experiments and analyses of many new structures proved the cluster encoding demethoxyviridin biosynthesis, and identified several new biosynthetic intermediates. This resulted in a much better and more detailed biosynthetic pathway, including some unusual enzyme complexes. The manuscript is overall well written, except for some minor errors (e.g., line 204 should be ... did not yield stable intermediates or ... did not yield any stable intermediate), well illustrated and discussed. Especially the color coding used in the figures is very clear and clever. The history of the cluster identification as shown in the Supporting materials is a bit confusing, since the genes shown in SI-Figs. 1 (g3266 to g3283) 5 (g3262-g3284) and the final gene cluster shown in Fig. 1 (vidA to vidS) don't match exactly. In the experimental section, the space between the numeral value and the degree centigrade symbols are missing. The NMR data need to be overhauled, since the couplings constants (J values) do not match. I recommend acceptance after minor corrections.

Reviewer #3 (Remarks to the Author):

The manuscript authored by Wang and collaborators brings substantial evidences about the biosynthetic pathway of furanosteroidal bioactive compounds. The conclusions are supported by a series of elegant genetic and chemical experiments, which led to the establishment of the biosynthetic pathway for this series of bioactive steroidal natural products. The manuscript is clearly written, and both main text and supplementary information contain technically sound genetic and physicochemical data that support their results and conclusions. The results are novel, adding consistent and interesting information in the literature for natural products scientists. So it could be accepted after some minor considerations:

The paper entitled "The viridin biosynthesis gene cluster of *Trichoderma virens* and its conservancy in the bat white-nose fungus *Pseudogymnoascus destructans*" has been recently published by Bansal et al. in *Chemistry Select* 2018, 3: 1289-1293. So, this paper appears to be the first genetic characterization of the BGC encoding for a demethoxyviridin related compound. Authors should consider including some connection between their results and the previously published paper.

Introduction, p. 3 line 47: Authors state that "Steroids are organic compounds possessing a seventeen-carbon tetracyclic ring system". Indeed, the tetracyclic system contains seventeen carbons; however, extra carbons (methyl groups) are part of the steroidal nucleus. I suggest rephrasing this sentence, considering that steroids are modified triterpenoids containing the tetracyclic system of lanosterol, but lacking the three methyl groups at C-4 and C-14. Further modifications in the side chain lead to different sub-classes of steroids bearing C₁₈ to C₂₉ skeletons.

Results, p. 10 lines 266-267 and Conclusions, p. 12 lines 321-322: Authors state demethoxyviridin derivatives are potential candidates for cancer treatment and conclude that their findings could be used for combinatorial chemistry to expand the chemical diversity of furanosteroids for drug discovery. Indeed, their findings can be used in combinatorial biosynthetic experiments for structural diversification. However; I do not completely agree with the direct application in drug discovery and potential use for cancer treatment, since they have applied only in vitro assays in this study. Demethoxyviridin has stability issues in biological systems, and some semi-synthetic derivatives acting as pro-drugs have shown better stability (Yuan et al. *Bioorg. Med. Chem. Lett.* 2009, 19: 4223-4227 cited in the manuscript as ref. 36). Demethoxyviridin and wortmannin also inhibit phospholipase A2 (Cross et al. *J. Biol. Chem.* 1995, 270: 25352-25355). Therefore, to state an application in drug discovery, these issues should be considered. To me, the main goal of this manuscript is the characterization of demethoxyviridin biosynthetic

pathway.

Supplementary Data: please check some constant coupling (J) values between protons coupled to each other, which should be the same. For example: in compound 2 (Supplementary Table 6) $J_{\text{H2a-H2b}} = 13.5$ Hz while $J_{\text{H2b-H2a}} = 13.3$ Hz; $J_{\text{H1-H2b}} = 11.7$ Hz while $J_{\text{H2b-H1}} = 11.6$ Hz. Please verify for other compounds.

Responsive letter to reviewers

REVIEWERS' COMMENTS:

Reviewer #1 (Remarks to the Author):

The manuscript entitled “Biosynthetic Pathway for Furanosteroid Antibiotic Demethoxyviridin and Identification of an unusual Pregnane Side-chain Cleavage” by Wang et al describes the sequencing of the demethoxyviridin biosynthetic gene cluster from the fungus *Nodulisporium* sp No 65-71-2-1, the confirmation that the gene cluster encodes the natural products via loss of function mutagenesis, dissection of the late stages of the biosynthetic pathway via systematic deletion of individual biosynthetic genes and characterization of the newly formed intermediates, heterologous expression of the early stages of the gene cluster in *Aspergillus oryzae* NSAR1 confirming the genes responsible for the early stage of the biosynthesis, enzymatic assay with recombinant purified BVFMO unambiguously demonstrating that this enzyme is responsible for side-chain cleavage, and heterologous expression of the side-chain cleavage three gene cluster in *A. oryzae* demonstrating its substrate tolerance.

This work clearly identifies the biosynthetic pathway for formation of demethoxyviridin and by extension, likely many of the related furanosteroids. While these compounds have been studied extensively, identification of the biosynthetic gene cluster has eluded researchers. Wang and coworkers used a new approach for identification of the gene cluster. Typically researchers mine genomes for isoprenoid pathways by looking for a core terpene cyclase gene. Because furanosteroids are generated from lanosterol, which is produced via the main primary steroid biosynthetic pathway and not a terpene cyclase, this approach could not identify the pathway. Wang and coworkers instead mined the genome for clustered P450, which are required for the significant oxidative processing of the compound. This enabled them to rapidly find putative gene clusters, evaluate them based on transcription and unambiguously identify the pathway via loss of function mutagenesis. This is an important addition to the strategies used to identify gene clusters in fungi.

The authors clearly deduce the order of the steps in the biosynthesis and the role of many of the genes responsible for demethoxyviridin production. This work will have a significant impact on the field. They show that C ring aromatization occurs similarly to A ring aromatization in estradiol biosynthesis and that loss of the pregnane side-chain occurs in this scaffold occurs via a BVFMO as is known for loss of the sidechain from progesterone in fungi.

Lastly the authors isolated and convincingly characterize 14 intermediates, 8 of which are new compounds. Using these compounds they are able to add to the structure activity relationship for the viridian class of PI3K inhibitors.

This work will impact researchers far beyond the biosynthesis community. This manuscript should be published in Nature Communications following minor revisions.

Response: we appreciate the reviewer's kind comments

1. Line 151. C10 demethylation, rather than C14, is more appropriate comparison since it likewise generates an aromatic ring. Please adjust.

Response: Thanks a lot for your notice. As the reviewer suggested, it is more appropriate to compare C13 demethylation with C10 demethylation in estrogen biosynthesis since both are involved in aromatic ring formation. We have modified the sentence as "VidA is responsible for the removal of the C18 methyl group before the action of VidK, a process analogous to the removal of the methyl group at C10 during estrogen biosynthesis." and added a corresponding reference 32 (page 6, lines 154-156).

2. Lines 239-242. The authors indicate that this experiment demonstrates a "conserved sterol metabolic pathway". It does not. BY demonstrating that the viridian biosynthetic proteins can also process progesterone, the authors have demonstrated that the enzymes have some degree of substrate tolerance. To determine if this is a conserved pathway, that is the pathway is present in many and diverse fungal genomes, the authors would need to mine multiple genomes and confirm that vidF, vidH, and vidP orthologs are present. Please correct.

Response: We agree with the reviewer. The conclusion we drew here is inappropriate. According the reviewer's suggestion, we have deleted the description "suggesting that the three enzyme mediated pregnane side chain cleavage is a conserved sterol metabolic pathway in the fungal genome" (page 9, line 244). However, using MultiGeneBlast, we really found the presence of VidF, VidP, and VidH orthologs in several fungal genomes (Supplementary Fig. 10). Considering the fact that fungal degradation of progesterone has been reported in a lot of fungi, we thus speculated that the three enzyme mediated pregnane side chain cleavage is likely to be a conserved sterol metabolic pathway in the fungal genome. These issues have been addressed in the discussion part (page 11, lines 307-308).

Reviewer #2 (Remarks to the Author):

This is excellent work by Abe, Gao, and co-workers. An elegant screening was performed to find the first gene cluster of furanosateroids. Gene inactivation experiments and analyses of many new structures proved the cluster encoding demethoxyviridin biosynthesis, and identified several new biosynthetic intermediates. This resulted in a much better and more detailed biosynthetic pathway, including some unusual enzyme complexes.

Response: we are grateful for the reviewer's kind comments

1. The manuscript is overall well written, except for some minor errors (e.g., line 204 should

be ... did not yield stable intermediates or ... did not yield any stable intermediate), well illustrated and discussed. Especially the color coding used in the figures is very clear and clever.

Response: Thanks a lot for the reviewer's notice. We have corrected the description "did not yield any stable intermediates" as "did not yield stable intermediates" (page 8, line 208).

2. The history of the cluster identification as shown in the Supporting materials is a bit confusing, since the genes shown in SI-Figs. 1 (g3266 to g3283) 5 (g3262-g3284) and the final gene cluster shown in Fig. 1 (vidA to vidS) don't match exactly.

Response: As the reviewer suggested, the description about the history of the cluster identification shown in the Supporting materials is a little confusing. To be consistent with Fig.1 (vidA to vidS), we have added additional information in the supplementary Figures 1, and 3-6, which are highlighted by yellow color in SI.

3. In the experimental section, the space between the numeral value and the degree centigrade symbols are missing.

Response: We have added the space between the numeral value and the degree centigrade symbols throughout the manuscript and the Supporting materials.

4. The NMR data need to be overhauled, since the couplings constants (J values) do not match. I recommend acceptance after minor corrections.

Response: we have carefully checked all the NMR data including couplings constants in the Supporting materials and corrected these errors. These modifications are highlighted by yellow color.

Reviewer #3 (Remarks to the Author):

The manuscript authored by Wang and collaborators brings substantial evidences about the biosynthetic pathway of furanosteroidal bioactive compounds. The conclusions are supported by a series of elegant genetic and chemical experiments, which led to the establishment of the biosynthetic pathway for this series of bioactive steroidal natural products. The manuscript is clearly written, and both main text and supplementary information contain technically sound genetic and physicochemical data that support their results and conclusions. The results are novel, adding consistent and interesting information in the literature for natural products scientists. So it could be accepted after some minor considerations:

Response: Thanks a lot for the reviewer's kind comments on our manuscript.

1. The paper entitled “The viridin biosynthesis gene cluster of *Trichoderma virens* and its conservancy in the bat white-nose fungus *Pseudogymnoascus destructans*” has been recently published by Bansal et al. in *Chemistry Select* 2018, 3: 1289-1293. So, this paper appears to be the first genetic characterization of the BGC encoding for a demethoxyviridin related compound. Authors should consider including some connection between their results and the previously published paper.

Response: Thanks for the reviewer’s notice and we apologized for our negligence of this paper. We have added some description on this paper in the introduction section and cited this paper in the main text (reference 22). Though this paper might be the first report on the gene cluster of viridin, the author did not provide substantial evidence for the biosynthetic pathway of viridin (page 4, lines 81-83).

2. Introduction, p. 3 line 47: Authors state that “Steroids are organic compounds possessing a seventeen-carbon tetracyclic ring system”. Indeed, the tetracyclic system contains seventeen carbons; however, extra carbons (methyl groups) are part of the steroidal nucleus. I suggest rephrasing this sentence, considering that steroids are modified triterpenoids containing the tetracyclic system of lanosterol, but lacking the three methyl groups at C-4 and C-14. Further modifications in the side chain lead to different sub-classes of steroids bearing C18 to C29 skeletons.

Response: We appreciate the reviewer’s constructive suggestion. According to the reviewer’s suggestion, we have rephrased the sentence “Steroids are organic compounds possessing a seventeen-carbon tetracyclic ring system” as “Steroids are modified triterpenoids containing the tetracyclic system of lanosterol, but lacking the three methyl groups at C4 and C14. Further modifications in the side chain lead to different sub-classes of steroids bearing C₁₈ to C₂₉ skeletons” (page 3, lines 48-50).

3. Results, p. 10 lines 266-267 and Conclusions, p. 12 lines 321-322: Authors state demethoxyviridin derivatives are potential candidates for cancer treatment and conclude that their findings could be used for combinatorial chemistry to expand the chemical diversity of furanosteroids for drug discovery. Indeed, their findings can be used in combinatorial biosynthetic experiments for structural diversification. However; I do not completely agree with the direct application in drug discovery and potential use for cancer treatment, since they have applied only in vitro assays in this study. Demethoxyviridin has stability issues in biological systems, and some semi-synthetic derivatives acting as pro-drugs have shown better stability (Yuan et al. *Bioorg. Med. Chem. Lett.* 2009, 19: 4223-4227 cited in the manuscript as ref. 36). Demethoxyviridin and wortmannin also inhibit phospholipase A2 (Cross et al. *J. Biol. Chem.* 1995, 270: 25352-25355). Therefore, to state an application in drug discovery, these issues should be considered. To me, the main goal of this manuscript is the characterization of demethoxyviridin biosynthetic pathway.

Response: We apologized for overselling our results. We have deleted the description on drug discovery or cancer treatment. (Page 10, lines 267-268 and Page 12, line 325).

4. Supplementary Data: please check some constant coupling (J) values between protons coupled to each other, which should be the same. For example: in compound 2 (Supplementary Table 6) $J_{H2a-H2b} = 13.5$ Hz while $J_{H2b-H2a} = 13.3$ Hz; $J_{H1-H2b} = 11.7$ Hz while $J_{H2b-H1} = 11.6$ Hz. Please verify for other compounds.

Response: we have carefully checked all the NMR data including couplings constants in the Supporting materials and corrected these errors. These modifications are highlighted by yellow color.

REVIEWERS' COMMENTS:

Reviewer #1 (Remarks to the Author):

The revised manuscript adequately addresses the reviewers' comments. This manuscript is now appropriate for publication in Nature Communications.

Reviewer #2 (Remarks to the Author):

This OK now.

Reviewer #3 (Remarks to the Author):

Authors have properly addressed the comments and suggestions. The manuscript can be accepted for publication

Responsive letter

REVIEWERS' COMMENTS:

Reviewer #1 (Remarks to the Author):

The revised manuscript adequately addresses the reviewers' comments. This manuscript is now appropriate for publication in Nature Communications.

Response: **Thanks a lot for the reviewer's kind comments.**

Reviewer #2 (Remarks to the Author):

This OK now.

Response: **Thanks a lot for the reviewer's kind comments.**

Reviewer #3 (Remarks to the Author):

Authors have properly addressed the comments and suggestions. The manuscript can be accepted for publication

Response: **Thanks a lot for the reviewer's kind comments.**